EMBO
Molecular Medicine

# Lowering mutant huntingtin by small molecules relieves Huntington's disease symptoms and progression

Anat Bahat [1,✉], Elad Itzhaki [1], Benjamin Weiss [1], Michael Tolmasov[2], Michael Tsoory[3], Yael Kuperman [3], Alexander Brandis[4], Khriesto A Shurrush[5] & Rivka Dikstein [1,✉]

## Abstract

Huntington's disease (HD) is an incurable inherited disorder caused by a repeated expansion of glutamines in the huntingtin gene (*Htt*). The mutant protein causes neuronal degeneration leading to severe motor and psychological symptoms. Selective downregulation of the mutant *Htt* gene expression is considered the most promising therapeutic approach for HD. We report the identification of small molecule inhibitors of Spt5-Pol II, SPI-24 and SPI-77, which selectively lower mutant *Htt* mRNA and protein levels in HD cells. In the BACHD mouse model, their direct delivery to the striatum diminished mutant Htt levels, ameliorated mitochondrial dysfunction, restored BDNF expression, and improved motor and anxiety-like phenotypes. Pharmacokinetic studies revealed that these SPIs pass the blood-brain-barrier. Prolonged subcutaneous injection or oral administration to early-stage mice significantly delayed disease deterioration. SPI-24 long-term treatment had no side effects or global changes in gene expression. Thus, lowering mutant Htt levels by small molecules can be an effective therapeutic strategy for HD.

**Keywords** DSIF; Huntington's Disease; RNA Pol II; Spt5; Spt5-Pol II inhibitor
**Subject Category** Neuroscience

## Introduction

Huntington's disease (HD) is a devastating genetic disorder caused by an expansion of trinucleotide repeat (CAG) in the first exon of the Huntingtin gene (*Htt*). The translated mutant huntingtin protein contains a polyglutamine (PolyQ) stretch that causes progressive degeneration of nerve cells in the brain and dramatically affects the functional and cognitive abilities of the patient (Gil

and Rego, 2008). Evidence for additional molecular mechanisms that contribute to HD toxicity include loss-of-function wild-type HTT protein (Cattaneo et al, 2001, 2005), and RNA toxic gain-of-function (Schilling et al, 2019; Sun et al, 2015).

Currently there are no effective treatments to delay the onset or slow the progression of the disease and lowering the levels of the *Htt* gene is among the most promising therapeutic strategies for HD. Several approaches were applied to lower *Htt* levels, mainly by administration of Htt antisense oligonucleotides (ASOs) (Kordasiewicz et al, 2012; Rook and Southwell, 2022). However, recent clinical trials with ASOs were halted (Kingwell, 2021). Their failure was attributed to treatment of patients at too advanced disease stages, too high dosing of the drugs and the potential to reduce the expression of the wt protein. Another effective approach for HD therapy would be the selective decreasing of the mutant *Htt* with minimal effect on the wt gene product by using small molecules that can pass the blood-brain barrier (BBB).

Transcription elongation by RNA polymerase II (Pol II) is a highly controlled process involving multiple transcription elongation factors, among them the Spt4/Spt5 complex. The large Spt5 subunit (p160) interacts with the small subunit Spt4 (p14) to form the DSIF complex (Wada et al, 1998; Yamaguchi et al, 1999) that has positive and negative effects on transcription, particularly of pro-inflammatory genes (Ainbinder et al, 2004; Amir-Zilberstein et al, 2007; Diamant et al, 2012; Diamant et al, 2016a; Diamant and Dikstein, 2013). Spt5 has multiple interaction domains that directly bind with RNA Pol II to stabilize it during elongation, and contacting the DNA and the nascent RNA (Bernecky et al, 2016; Blythe et al, 2016; Hirtreiter et al, 2010; Klein et al, 2011; Martinez-Rucobo et al, 2011). Depletion of Spt5 in mammalian cells does not result in broad transcriptional effects (Diamant et al, 2012, 2016a; Fitz et al, 2018; Komori et al, 2009; Pavri et al, 2010; Stanlie et al, 2012), suggesting a redundancy among mammalian transcription elongation factors. While Spt5 is central to pro-inflammatory gene regulation, Spt4 is dispensable (Diamant et al, 2016a). Several studies have shown that Spt4 and Spt5 have a specific role in transcribing genes with abnormal long stretches of multiple repeats. This includes the mutant *Htt*

[1]Department of Biomolecular Sciences, The Weizmann Institute of Science, Rehovot 76100, Israel. [2]The Mina & Everard Goodman Faculty of Life-Sciences and The Leslie & Susan Gonda Multidisciplinary Brain Research Center Bar-Ilan University, Ramat-Gan 5290002, Israel. [3]Department of Veterinary Resources, Weizmann Institute of Science, Rehovot 76100, Israel. [4]Life Sciences Core Facilities, Weizmann Institute of Science, Rehovot 76100, Israel. [5]The Nancy and Stephen Grand Israel National Center for Personalized Medicine, The Weizmann Institute of Science, Rehovot 76100, Israel. ✉E-mail: anat.bahat@weizmann.ac.il; rivka.dikstein@weizmann.ac.il

associated with HD, *C9orf72* of amyotrophic lateral sclerosis and frontotemporal dementia (ALS/FTD) and *NOP56* of spinocerebellar atrophy type 36 (SCA36) (Liu et al, 2012; Cheng et al, 2015; Kramer et al, 2016; Furuta et al, 2019). Interference with Spt4 or Spt5 by siRNA decreased the transcription of the above-mentioned mutant gene alleles containing long expansions but had little effect on the expression of genes containing normal repeat length. The requirement of Spt4/Spt5 for selective transcription of three distinct expanded repeat genes suggests a common mechanism that is presently unknown. These findings place Spt4/Spt5 as an attractive target for drugs against HD, ALS/FTD, SCA36, and potentially other repeat expansion neurodegenerative disorders.

We previously targeted the Spt5-Pol II complex in a high throughput drug screen using a protein-protein interaction assay (Ashkenazi et al, 2017) which led to the identification of the first Spt5-Pol II inhibitors called SPIs (Bahat et al, 2019). The identified SPIs precisely mimic the effect of Spt5 depletion on basal and activated pro-inflammatory genes. Similar to the Spt4 and Spt5 knockdown effects, several SPIs display a selective inhibition of mutant but not wt *Htt* gene transcription. Notably, a subset of SPIs was found to be selective for a single activity of Spt5, suggesting that the various activities of this factor can be uncoupled and selectively targeted by different inhibitors (Bahat et al, 2019). Inhibition of Spt4-Spt5 by 6-azauridine was also reported to affect the expression of the mutant *Htt* allele (Deng et al, 2022). CRISPR editing of Spt4 in iPCS neuronal progenitors, which were transplanted into HD animals also reduces mutant HTT expression (Park et al, 2022). Together these findings reinforce the Spt4/Spt5 and Spt5-Pol II complexes as excellent targets for HD therapy.

Here we describe the development of two novel SPI analogs, SPI-24 and SPI-77, displaying mutant *Htt*-specific effects for HD therapy. Using the established BACHD mouse model of HD, we demonstrate the ability of SPI-24 and SPI-77 to pass the blood-brain barrier and to lower mutant *Htt* mRNA and protein levels in the striatum. SPI-treated BACHD mice show enhancement of mitochondrial DNA in the plasma and BDNF in the striatum, known to be reduced by mutant HTT. Furthermore, behavioral tests reveal amelioration of motor impairments and restoration of anxiety symptoms to WT levels, delaying disease progression. SPI-24 does not cause changes in global gene expression or notable side effects. Thus, SPIs can be considered excellent candidates for HD management and potential therapy.

# Results

## Identification of SPI analogs with improved potency

Our previous study identified several SPIs, including SPI-85, SPI-09, and SPI-31 that selectively inhibit transcription of mutant *Htt* without affecting WT *Htt* in striatal cell lines (Bahat et al, 2019). We selected these molecules for further study since their effect on the other known Spt4/Spt5 activities at the same effective concentration was minor and therefore considered as mutant *Htt*-specific inhibitors. To improve the potency of these compounds, we performed structure–activity relationship (SAR) by-catalog and identified 16 analogs structurally similar to SPI-85, SPI-09, and SPI-31. In addition, a chemical synthesis was performed to generate a new analog of SPI-31, SPI-4516, to reduce potential

reactivity (Appendix Fig. S1). To test the effect of these 17 analogs on mutant *Htt* expression, we treated with these compounds the Q111 striatal cells expressing a mutant *Htt* gene with 111 CAG repeats (Trettel et al, 2000) for 48 h, and determined the level of mutant *Htt* mRNA by RT-qPCR. The results, shown in Fig. 1A, indicate that several analogs can significantly inhibit the mutant *Htt* mRNA levels at concentrations much below those of the original molecules from which they were derived. Since SPI-09 and its analogs were suspected to react with other molecules in the cells, their analysis was discontinued. We next determined the half-maximal inhibitory concentration ($IC_{50}$) of the most effective analogs and found it to be in the low μM and nM range (Fig. EV1A). Based on the $IC_{50}$ and the chemical structure and reactivity, we selected three potent inhibitors for further analysis: SPI-1477, 4516, and 0324 (Fig. 1A right). We next validated that these improved analogs are selective against the mutant *Htt* by comparing their effect on the WT (Q7) and mutant *Htt* mRNA (Fig. 1B). As we aimed to examine the effect of prolonged treatment by these compounds, we tested their stability in saline solution for 28 days at 37 °C using LC-MS/MS. The analysis showed that only SPI-1477 (SPI-77) and SPI-0324 (SPI-24) were relatively stable in the solution for a long time (Table EV1), and therefore they were selected for further study (Fig. 1A right). Metabolic labeling of Q111 cellular RNA with 4-thiouridine (4sU) for 2 h in the presence of SPI-24 and SPI-77 revealed that the transcription of newly synthesized mutant *Htt* was diminished by these analogs (Fig. 1C), confirming that they act at the transcription level. We also analyzed the effect of the SPIs on the newly synthesized mRNA levels of other genes, which normally bear CAG repeats and found that non were reduced by the SPIs (Fig. EV1B). The differential reduction in the mRNA levels of mutant *Htt* compared to wt is also reflected in a decrease in the mutant protein level (Fig. 1D). Next, we treated with SPIs HD patient-derived heterozygous cells encoding wt and mutant *Htt* alleles bearing 44, 55, 66 and 180 CAG repeats. We found significant reduction of the mutant protein (detected by anti-polyQ antibody) while the effects on the total Htt (Fig. 1E,F) and wt Htt (Fig. 1G) were non significant.

## SPI-24 and SPI-77 selectively reduce mutant Htt in vivo

To evaluate the therapeutic potential of the potent SPIs *in vivo*, the well-characterized HD mouse model BACHD was used (Gray et al, 2008). This model was chosen for two main reasons. First, it expresses the full-length human mutant *Htt* gene containing 97 mixed CAG-CAA repeats. Second, the disease progress in BACHD is relatively slow, with symptoms developing gradually, starting approximately at 8–12 weeks of age and becoming more pronounced at 6 months (Menalled et al, 2009), resembling disease development in humans. At the same time, it has a robust phenotype and is, therefore, suitable for drug testing.

Mutant BACHD mice of 7–9 months age that already showed strong phenotype (Appendix Table S1) were treated with either vehicle (DMSO) or SPI-24 and SPI-77 by direct injection into the striatum, the brain region, which is mostly affected in HD (Fig. 2A). This was conducted using a metal infusion cannula connected to an osmotic minipump (Fig. 2B) that constantly infused the SPIs for 28 days. The tip of the cannula was stereotactically placed in the right brain striatum at specific coordinates (Perucho et al, 2013) (Fig. EV2A). The subcutaneously implanted pumps were pre-filled

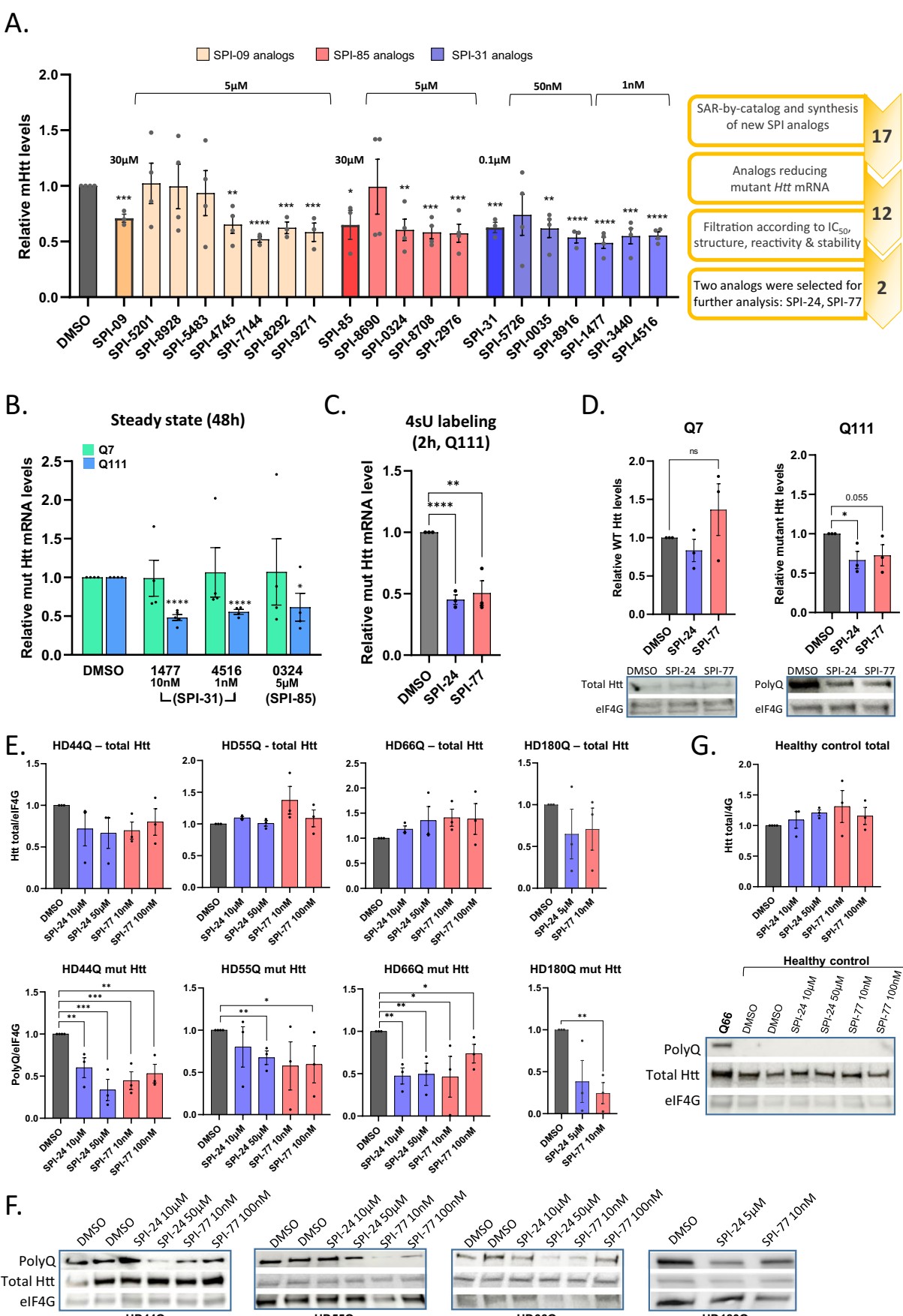

**Figure 1. Identification of potent SPIs analogs and their mechanism of action.**

(A) Q111 cells were treated with 17 analogs of 3 mutant *Htt* selective SPIs for 48 h. Then, RNA was extracted, and the levels of mutant *Htt* mRNA were determined by qRT-PCR and normalized to beta-actin. Each bar represents the means ± SEMs of 3–6 independent experiments. On the right: a scheme describing the steps that lead to the selection of SPI-24 and SPI-77. (B) Q111 and Q7 cells were treated with the 3 selected SPIs analogs for 48 h, and then the levels of *Htt* mRNA were determined by qRT-PCR. Each bar represents the means ± SEMs of 3–4 independent experiments. (C) Q111 cells were metabolically labeled with 4-thiouridine for 2 h in the presence of DMSO, SPI-24 (5 µM) or SPI-77 (10 nM). Newly synthesized RNAs were purified, and mRNA levels were determined by qRT-PCR. Each bar represents the means ± SEMs of 3 independent experiments. (D) Western blot analysis of Q7 and Q111 cells treated with DMSO, SPI-24 (5 µM) or SPI-77 (10 nM) for 48 h. Wt and mutant Htt were detected using anti-Htt and anti-PolyQ antibodies, respectively. Anti-eIF4G served as normalized protein. Each bar represents the means ± SEMs of 3 independent experiments. Representative blots are shown below the graphs. (E) Quantification of Western blot analysis of HD patient-derived cells bearing 44, 55, 66, and 180 repeats showing the level of mutant (anti-polyQ) and total HTT proteins relative to normalized protein (eIF4G1). Each bar represents the means ± SEMs of 3–4 independent experiments. (F) Representative blots of the data shown in panel (E). (G) Western blot analysis of normal, healthy, human cells as a control for the effect of SPIs on the wt protein and of HD66Q as a control for the polyQ antibody. Each bar represents the means ± SEMs of 3–4 independent experiments. Data information: The asterisks in panels (A–E) and (G) denote statistical significance differences relative to DMSO according to Student's unpaired *t*-test (one-tailed). *$p < 0.05$; **$p < 0.01$; ***$p < 0.005$; ****$p < 0.001$; ns, not significant. Source data are available online for this figure.

with either DMSO (25% in saline) or SPI-77 or SPI-24 at a concentration of 0.5 mM (0.6 µg/day) and 5 mM (5 µg/day), respectively. The disease progression was monitored by well-established behavioral tests conducted one week before the stereotactic surgery and again one week before sacrificing the mice (see study design scheme in Fig. 2A). In addition, blood samples were collected one week before the surgery and on the last day. Following 28 days of striatal infusion, the mice were sacrificed, brains were removed, sliced, and small punch-size pieces of tissue from the striatum (Fig. EV2B) were taken for RNA and protein analyses. The levels of wt (mouse) and mutant (human) *Htt* mRNA in the injected (right hemisphere) and non-injected (left hemisphere) striatum were analyzed using species-specific sets of primers by RT-qPCR. We found that in the injected striatum, both SPI-24 and SPI-77 significantly reduced the mut/wt mRNA ratio relative to DMSO treatment (Fig. 2C, left; *n* = 8 in each group). On the non-injected hemisphere, the reduction is less pronounced and non-significant, due to low diffusion to the non-injected hemisphere, as can be seen from the residual amount of SPI-24 in the injected vs non-injected hemisphere (Fig. EV2C). We performed immunoblot analysis on samples from the injected striatum of the treated mice. Total Htt (human and mouse) was detected using an anti-Htt antibody and mutant Htt was detected using polyQ antibody. The ratio between polyQ and total Htt was significantly reduced upon treatment with both SPIs (Fig. 2D).

We then evaluated the effect of the SPIs on the levels of two HD biomarkers. The mRNA level of the Brain-derived neurotrophic factor (BDNF), whose expression is known to be reduced in HD (Gutierrez et al, 2019; Zuccato et al, 2001) was analyzed from the striatal samples. The data presented in Fig. 2E confirmed the reduced BDNF levels in the BACHD mice (untreated WT vs DMSO-treated BACHD bars in Fig. 2E) and indicated that its mRNA levels were elevated following SPIs treatment, particularly SPI-24 whose effect was significant, while the *P*-value for SPI-77 was 0.06. HD is also associated with impaired mitochondrial integrity and excessive mitochondrial fission (Hwang et al, 2015) resulting in a decline in the level of the mtDNA in the brain and plasma of HD patients (by >50% in severe HD patients), as well as in HD mice model (Guo et al, 2013; Disatnik et al, 2016). We determined the transcript level of the mitochondrial gene mtND2 in the plasma of BACHD mice and found it to be reduced by 53% compared to the plasma of WT mice (Fig. 2F), similar to previous reports. The level of mtND2 of the post-treated mice was significantly higher than that of the pre-treated mice in both

SPIs-treated groups, in contrast to the DMSO group in which the level continued to drop (Fig. 2F), as expected from the progression of the disease without treatment. Thus, the effect of SPIs on HD biomarkers is consistent with the inhibitory effect of the SPIs on mutant *Htt* expression.

## SPIs attenuate HD-related behavioral phenotypes

To assess the effect of the SPIs on basal locomotor activity, exploratory activity, and anxiety-related behavior, we performed the open field test. In this test, we evaluated the total distance traveled by the mice in the arena, the frequency of transition to the center and the time spent in the center. No significant differences in all the 3 parameters were observed between WT mice and the BACHD mice in our setup, indicating no phenotypic difference (Appendix Fig. S2). Therefore, we conducted two alternative tests: the beam walk test, which assesses motor coordination and balance, and the Acoustic Startle Response (ASR) test to assess the sensorimotor functions and the anxious behavior of the mice (Brooks and Dunnett, 2009). In the beam walk assay, the mice have to cross a narrow beam that leads them to their empty home cage for 5 consecutive times. Performance on the beam is quantified by measuring their crossing time, the number of their paw steps while traversing the beam, and the percentage of their paw step slips (slip ratio, hereafter). Strong HD phenotype is reflected in a higher number of paw steps and slip ratio. Figure 2G–I presents the number of paw steps, the slip ratio, and the crossing time. BACHD DMSO-treated mice had a significantly higher number of paw steps and slip ratio compared to untreated WT mice, reflecting a strong HD phenotype. On the other hand, BACHD SPI-24-treated mice walked faster (3.3 ± 0.3 sec relative to 2.6 ± 0.2 s in DMSO; *P* = 0.01; Fig. 2I), with significantly fewer paw steps: the mean number of steps of the post-treated SPI-24 group is similar to that of the WT mice (Fig. 2G). Importantly, they slipped much less relative to pre-treatment and relative to the DMSO-treated group (Fig. 2H and Table 1). SPI-77 did not improve the motor coordination and balance of the mice.

To test whether the SPIs can affect HD-associated anxiety, we carried out the ASR assay. The experiment is comprised of three Blocks (B1–3). In the first (B1) and the third (B3) blocks, 6 stimuli of 120 decibels (dB) are generated, and the response of the mice to these stimuli is measured as a function of force (g). In general, healthy animals become less responsive to the repeated stimulus in B3 due to habituation, while anxious animals react the same in B1

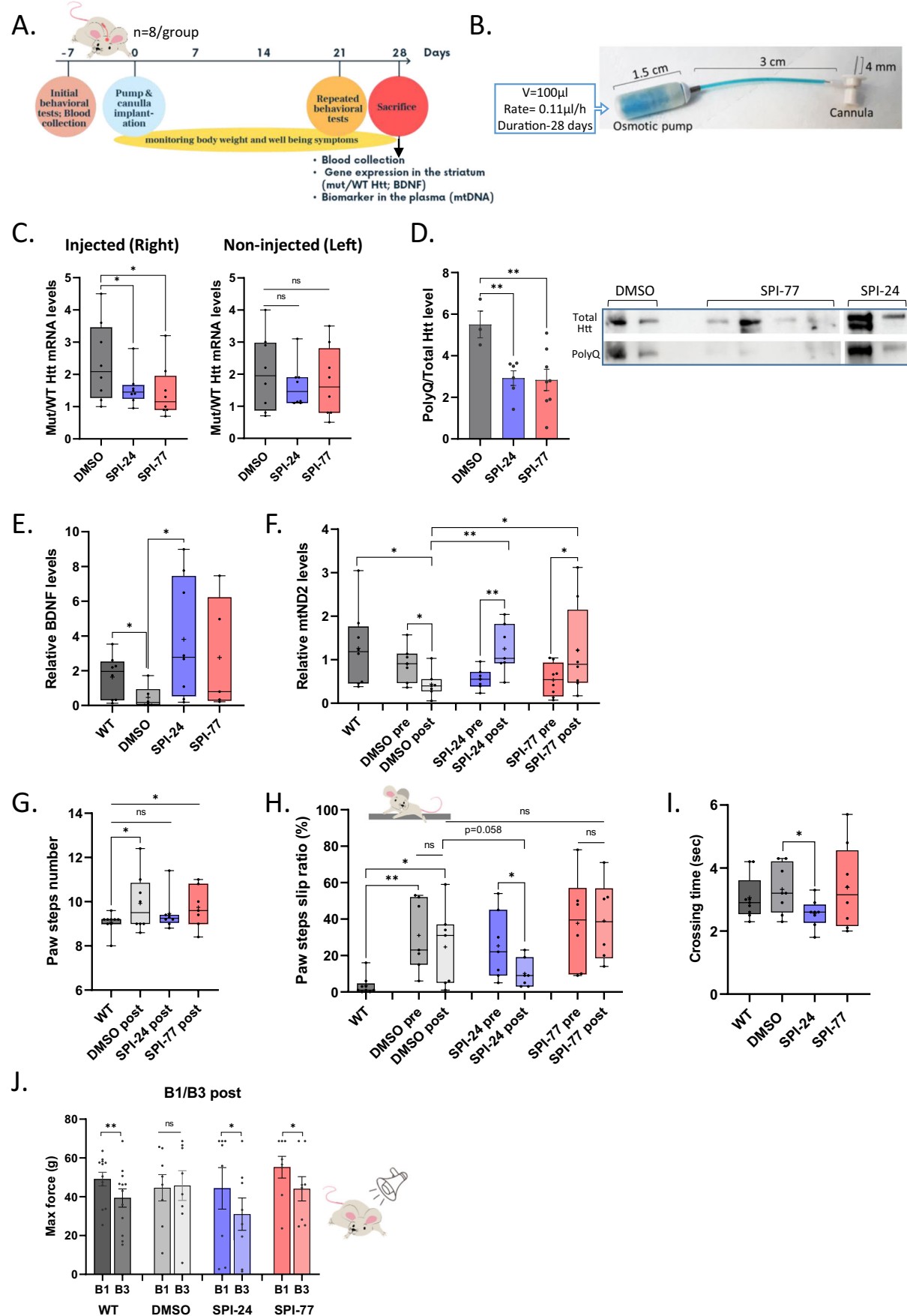

**Figure 2. Direct injection of SPIs to the striatum of BACHD mice.**

(A) The design of the striatum direct injection study using 7–9 months old BACHD mice. (B) Pre-filled ALZET osmotic pump connected to a metal infusion cannula by a short vinyl catheter. (C) The ratio of mutant and wt *Htt* mRNA in the injected and non-injected sides of the striatum of treated mice. The line represents the median of 8 mice. (D) Western blot quantitation showing the ratio of mutant (using anti-polyQ antibody) and total HTT protein (using anti-Htt antibody) in the injected striatum of treated mice. Each bar represents the means ± SEMs of 3–8 mice. A representative blot is shown on the right. (E) The relative levels of BDNF in the striatum of untreated WT and treated BACHD mice. The line represents the median of 7–8 mice. (F) The relative levels of mtND2 in the plasma of untreated WT and treated BACHD mice, before and after SPIs treatment. DNA was extracted from the plasma samples and assayed by qPCR. Beta-actin was used for normalization. The line represents the median of 7–8 mice. (G–I) Performance of untreated WT and treated BACHD mice in the Beam walk test, calculated as the mean number of paw steps (G), paw steps slip ratio (H) and crossing time (I). Each line represents the median of 7–8 mice. (J) The Acoustic startle response test: the mean force response (g) of untreated WT and treated BACHD mice to stimuli in B1 and B3 stages. Each bar represents the means ± SEMs of 7–8 mice. Data information: The lines and the '+' within the box-whisker plots (min to max) in panels (C), (E–I) represent the median and the average, accordingly. Asterisks denote statistically significant differences according to Student's unpaired one-tailed (in panels C, D, E, G and I) or paired two-tailed (in panels F, H and J) *t*-tests. Comparison to WT group in panels (F) and (H) was calculated as unpaired one-tailed *t*-test. The *t*-tests between WT and BACHD DMSO in panels (G) and (H) were corrected with Welch *t*-test since the populations had different standard deviations. *$p < 0.05$; **$p < 0.01$; ns, not significant. Source data are available online for this figure.

and B3 (Neufeld-Cohen et al, 2010). Figure 2J shows that while the response of untreated WT animals was reduced in B3, as expected, DMSO-treated BACHD animals did not habituate and displayed no change in their response. On the other hand, both SPIs-treated mutant animals reacted significantly less in B3 relative to B1, similar to the WT group, suggesting that the treatment was effective in reducing anxiety.

## Short-term subcutaneous injection or oral administration lowered mutant *Htt* expression

Encouraged by the results from the direct brain injection of SPIs, we next examined the potential effect of the SPIs by other administration routes. For this purpose, we first tested the stability of SPI-24 and SPI-77 in the mice plasma over time (pharmacokinetics) as well as their penetration ability through the blood-brain-barrier (BBB). Each compound was subcutaneously injected into WT mice at 2 different concentrations and blood samples were taken at 8 time points after injection (3 mice for each concentration and each time point; a total of 24 mice for each compound). After the last bleeding (12 h), the mice were sacrificed and their brain was analyzed (Fig. 3A). Three mice were left untreated (control) for background levels of the compounds in the blood and brain. All samples were then extracted and SPI-24 and SPI-77 concentrations were analyzed using LC-MS/MS. At the high dose (10 mg/Kg), SPI-24 reached its maximal concentration in the plasma 4 h after the injection and then dropped (Fig. 3B). Twelve hours after the injection SPI-24 was detected in the animal's brain, indicating that this compound can pass the BBB (Fig. 3C). The Cmax found in the brain was 26 μM, which is higher than the effective concentration (IC$_{50}$) of the compound. SPI-77 could not be detected using LC-MS/MS, even in a pilot experiment where it was spiked-in directly to the plasma or to the brain tissue. This is probably due to the presence of another endogenous compound that generates an identical peak that masks its signal. To test whether SPI-24 is stable in the brain longer than 12 h, we repeated the experiment with 10 mg/Kg SPI-24 and collected brains 12, 24, and 48 h later. This time, we also performed a brain perfusion to verify that the source of the detected compound was from the tissue itself and not from the blood vessels. The levels of SPI-24 after 24 and 48 h post-injection are much lower than after 12 h (Fig. EV3A), suggesting the decomposition of the compound over time.

Based on the pharmacokinetics experiments, we next tested the effect of a daily subcutaneous (SC) injection or oral administration

(gavage) of SPI-24 (SC: 38 and 190 μg/day; oral: 490 μg/day) and SPI-77 (SC: 4.4 and 22 μg/day; oral: 55 μg/day) on the expression levels of wt and mutant *Htt* mRNA in the striatum of BACHD mice (see Appendix Table S1 for mice list). The compounds were delivered daily for 4 subsequent days and 6 h after the last administration, the mice were sacrificed. As can be clearly seen in Fig. 3D,G, the mutant to wt *Htt* mRNA ratio in the striatum was significantly reduced upon the daily short-term treatment with both SPIs using SC and oral delivery methods. Remarkably, SPI-77 reduced the mutant *Htt* mRNA upon subcutaneous injection in a concentration as low as 2.2 μg/day (0.1 mM; Fig. 3D), indicating that it also can pass the BBB. Accordingly, the protein levels of mutant Htt were significantly reduced following SC injections (Fig. EV3B). The remaining concentration of SPI-24 in the mice's brain after short SC injection is similar to those found in the PK experiment (Fig. EV3C). The BDNF levels also showed a trend of elevation following these treatments (Fig. 3E,H). These results indicate that both SPIs can be delivered subcutaneously or orally, pass the BBB and reduce mutant *Htt* expression in the striatum. We injected a mixture of SPI-24 and SPI-77 but observed no additive or synergistic effect (Fig. EV3D), suggesting that both compounds, although chemically distinct, act in a similar manner.

To examine the global effect of short-term SPIs treatment, RNA samples from injected mice were subjected to RNA sequencing (RNA-seq). Reads were aligned to the mouse genome, and the normalized expression of each gene was determined. Remarkably, only 13 genes were upregulated and one gene was down-regulated in response to SPI-24 treatment (Fig. 3F and Appendix Table S2), suggesting no global effect caused by SPI-24. On the other hand, a total of 1106 genes displayed a statistically significant fold change upon SPI-77 treatment (928 genes were down-regulated and 178 were upregulated; Fig. 3F and Appendix Table S2).

We also examined the effect of SPIs on alternative splicing of selected genes previously reported to be modulated in HD (Elorza et al, 2021) and found no significant effect (Appendix Fig. S3).

## Long-term SPI-24 treatment of early-stage mice delays HD progression

In neurodegenerative diseases, such as HD, the time of intervention is critical. The destruction of the neurons is mostly irreversible, and restoring the neurological damage in HD patients is challenging. Therefore, early intervention at the prodromal stage, or even before as a preventive approach, is likely to be beneficial. For this reason,

**Table 1. Summary of the effect of the SPIs on the slip ratio index of the mice analyzed from the beam walk test following different delivery methods.**

| Delivery method | Age | Testing time | Paw slip ratio | | |
| --- | --- | --- | --- | --- | --- |
| | | | DMSO | SPI-24 | SPI-77 |
| Direct | 7 months | Pre | 31.0 ± 7.3% | 25.3 ± 7.1% | 37.7 ± 10.8% |
| | | Post (1 m) | 24.7 ± 8.1% | 10.1 ± 3.0% | 39.0 ± 9.1% |
| | | Disease progression | 1.8 ± 0.9 | 0.6 ± 0.2* | 1.3 ± 0.3 |
| Oral | 3 months | Pre | 4.6 ± 1.2% | 8.4 ± 1.8% | 6.0 ± 1.4% |
| | | Post (2 m) | 22.9 ± 4.5% | 18.1 ± 5.7% | 16.6 ± 6.0% |
| | | Disease progression | 5.0 ± 0.9 | 1.9 ± 0.4* | 4.1 ± 1.5 |
| SC pump | 3 months | Pre | 11.8 ± 2.7% | 10.5 ± 3.3% | Not tested |
| | | Post (2 m) | 30.5 ± 9.5% | 7.7 ± 3.4% | |
| | | Disease progression | 3.8 ± 2.0 | 0.9 ± 0.3* | |

The numbers are the means ± SEMs of the slip ratio (%). The disease progression index was calculated for each mouse by dividing the slip ratio post treatment by the ratio pre treatment.

Data information: Asterisks denote statistically significant differences relative to DMSO according to Student's paired one-tailed $t$-test. *$p < 0.05$.

*Statistically significant ($p < 0.05$) relative to DMSO according to Student's paired one-tailed $t$-test.

we initiated an experiment in which we tested the effect of the SPIs on disease progression in young, 12-week-old mice, an age known to display weak HD phenotype (Menalled et al, 2009). As a first attempt, we provided the BACHD mice with the compounds (SPI-24: 35 mg daily; SPI-77: 4 mg daily) in their drinking water (containing 8% fructose) for 2 months (see study design in Fig. 4A and Appendix Table S1). Water-containing DMSO/SPIs were refreshed every 3 days based on a preliminary experiment that tested the stability of the two compounds in similar conditions, which indicated a decrease over time in their stability levels (Table EV2A). The disease progression was monitored by 3 behavioral tests: ASR, beam walk test, and home-cage locomotion assessment. The tests were conducted one week before the treatment started and again one week before sacrificing the mice (after 8 weeks, Fig. 4A). The results obtained from the ASR test show no significant change in the B3/B1 ratio in the pre-treated BACHD compared to untreated WT animals (Fig. 4B left), suggesting that the mutant animals already display an anxious phenotype. On the other hand, in both SPIs-treated groups but not DMSO, there is a significant improvement in the habituation to the stimuli in B3 relative to B1 (Fig. 4B right), indicating restoration to WT phenotype.

The beam walk test in this experiment was designed so that the pre-treated mice had to cross a 10 mm beam while at the end of the treatment (7 weeks later) they had to cross a 5 mm beam to reduce the learning effect. The analyses of this test (Fig. 4C and Table 1) indicated a more pronounced increase in the slip ratio over time in the DMSO-treated group (5.0 ± 0.9 times) compared to the SPIs-treated groups (1.9 ± 0.4 times in SPI-24; $P = 0.0048$, and 4.1 ± 1.5 times in SPI-77). These results suggest a slowdown in developing impaired motor and balance symptoms following SPI-24 treatment.

An additional test we performed was the home-cage locomotion (HCL). In this test, the mice's movement was tracked and analyzed for 48 h in their home-cage. Comparing the activity of untreated WT mice to pre-treated BACHD mice revealed that BACHD mice are more hyperactive than the WT, mainly during the active, dark phase, reflecting their involuntary movement (Fig. 4D, top panel: BACHD vs WT). 7 weeks after the treatment started, the hyperactive movement of the DMSO-treated mice was further

elevated (Fig. 4D, BACHD DMSO panel). In contrast, treatment with SPI-24 and SPI-77 prevented the enhancement of this abnormal activity (Fig. 4D, two bottom panels and summary graphs in the right panels). Together, these findings demonstrate that treatment with SPIs inhibits the disease progression and SPI-24 seems more potent than SPI-77, most likely because it is more stable in high concentration. The remaining concentration of SPI-24 in the mice's brain after long-term oral administration is shown in Fig. EV4A. Analyzing the protein levels of mutant and wt Htt in treated mice revealed a significant reduction in mutant Htt levels in both SPIs-treated groups (Fig. EV4B), which is in line with the phenotypic improvement.

Considering our inability to control precisely the volume of the water-containing SPIs consumed by the mice, we designed another long-term treatment with the more stable SPI-24. In this trial, we constantly delivered the compound to early-stage HD mice using a subcutaneous osmotic pump. The pumps were pre-filled with either DMSO-PEG400 (vehicle, this solvent was chosen based on a formulation screen, see Table EV2B) or SPI-24 at a concentration of 80 mM (187 µg/day considering a constant rate of 0.25 µl/h), and implanted under the mice's skin for 56 days (Figs. 5A and EV5A). The pump was replaced once after 28 days. Behavioral tests were performed a week before the treatment started and 7 weeks later, just before it ended (see Fig. 5A for study design and Appendix Table S1). According to the Beam walk test, there was a significant aggravation in the DMSO-treated group ($n = 11$) over time, as expected without treatment (Fig. 5B, left and Table 1). On the other hand, the treatment with SPI-24 ($n = 11$) prevented the increase in the paw slip frequency (Fig. 5B, left, and Table 1). The climbing and the wheel running tests that were conducted to analyze the functional abilities of the treated-mice, also showed a significant improvement following SPI-24 treatment relative to DMSO, and the differences relative to untreated WT mice become insignificant (Fig. 5B, middle and right panels, respectively). The home-cage locomotion data indicated some elevation in hyperactivity after 7 weeks in DMSO-treated mice (mainly at D5–D10 period; Fig. 5C, upper left panel and right panel), however, a reduction in hyperactivity was evident in SPI-24 treated mice (Fig. 5C, lower left panel and right panel). Analysis of brain samples by LC-MS/MS

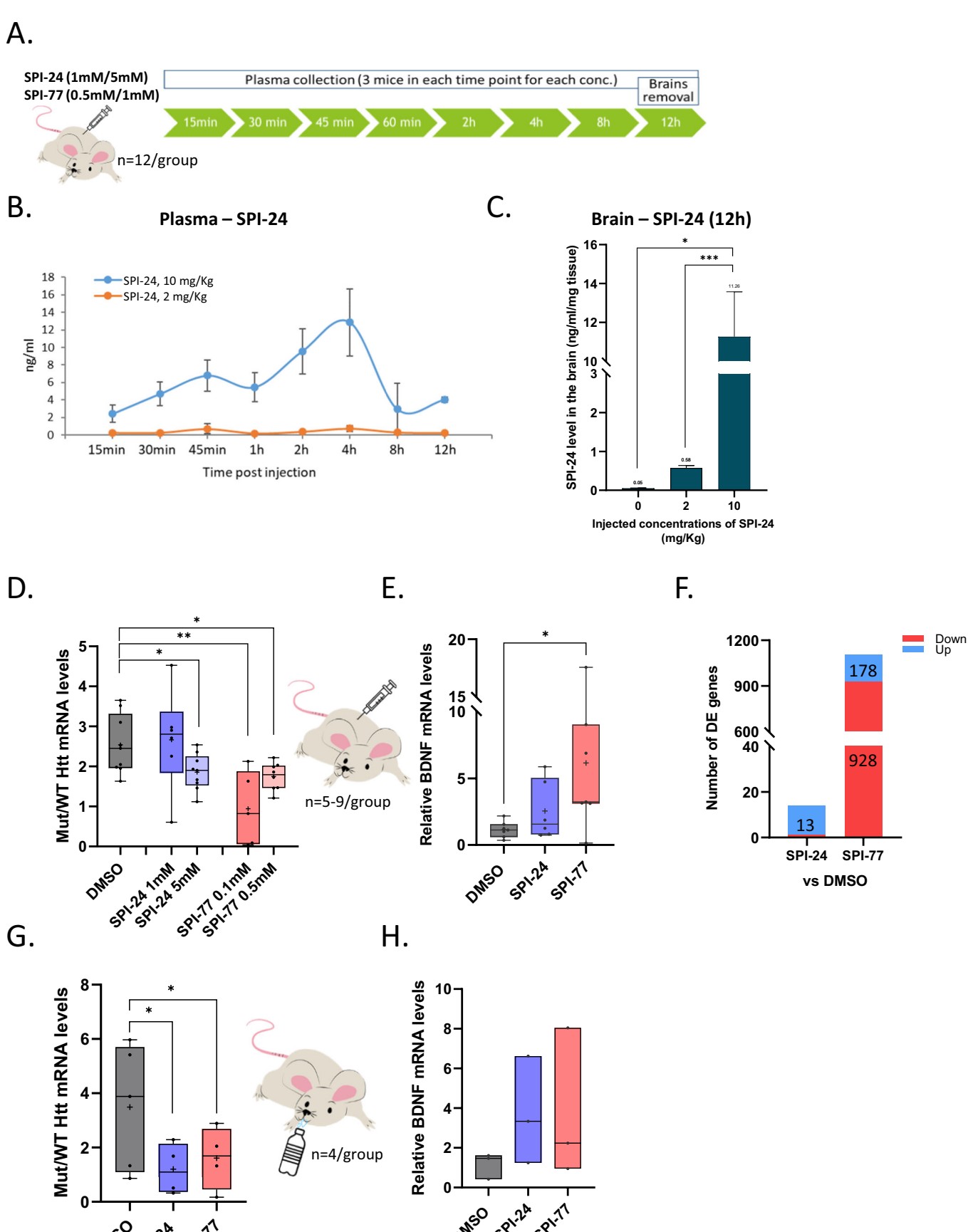

◄ **Figure 3.   Analysis of short-term subcutaneous injection or oral administration.**

(A) Schematic design of the pharmacokinetic experiment. (B) The concentration of SPI-24 in the mice plasma was determined using LC-MS/MS, calculated as ng/ml and normalized to total protein concentration. Each point is a mean ± SEM of 3 animals. (C) The remaining concentration of SPI-24 in the mice's brain 12 h after injection was determined using LC-MS/MS, calculated as ng/ml and normalized to tissue weight. Each bar represents the means ± SEMs of 12 animals. D, G The ratio of the levels of mutant and wt *Htt* mRNA in the striatum of treated mice upon subcutaneous injection (D, $n = 5$–9) and oral administration (G, $n = 4$–5) of the compounds. (E,H) The relative levels of BDNF in the striatum of subcutaneously (E) and orally (H) treated BACHD mice ($n = 3$–6). (F) The global effect of SPI-24 and SPI-77 on the number of differentially expressed genes, compared to DMSO ($n = 3$–5). Sequencing libraries from mRNA extracted from the striatum of injected mice were prepared and subjected to deep sequencing. Reads were aligned to the mouse genome. Data information: The lines and the '+' within the box-whisker plots (min to max) in panels (D), (E), (G) represent the median and the average, accordingly. The lines in the floating bars (min to max) in panel (H) represents the median. The asterisks in panel (C) indicate significant differences relative to naïve animals according to Student's unpaired one-tailed *t*-tests. The asterisks in panels (D, E, G and H) denote statistically significant differences relative to DMSO according to Student's unpaired two-tailed *t*-tests. *$p < 0.05$; **$p < 0.01$; ***$p < 0.005$. Source data are available online for this figure.

indicated that after 28 days, the residual concentration of SPI-24 is very low (Fig. EV5C), indicating a decline in its stability over a long period in the pump. This is in line with no detectable effect on the Htt protein level upon prolonged treatment with the pump (Fig. EV5D), and no improvement in the anxiety symptoms (Fig. EV5B).

We analyzed the global effect of the long-term SPI-24 treatment on gene expression by RNA-seq and found that only 42 genes were differentially expressed (39 genes were downregulated and 3 were upregulated, Fig. 5D and Appendix Table S3).

In addition to the effect of SPI-24 on the disease progression, we tested if there were any toxic effects on the mice following the prolonged treatment. For that purpose, we determined their water and food intake (Fig. 5E,F), their body composition (Fig. 5G) and the levels of different biochemical components in their blood (Fig. 5H) following 56 days of SPI-24 subcutaneous treatment. The results indicated no change in all the tested parameters following prolonged treatment.

## Discussion

In the current study, we demonstrate the remarkable efficacy of two Spt5-Pol II inhibitors in a mouse model of HD and provide support for the potential of selective lowering of mutant *Htt* by small molecules as an effective therapeutic approach for the treatment of HD-associated neuropathology. We previously identified a set of Spt5-Pol II small molecule inhibitors that selectively diminish the transcription of mutant but not wt *Htt* gene in striatal cell lines (Bahat et al, 2019). We now report the identification of potent SPI analogs that selectively reduce mutant *Htt* levels in the nM to low µM range. This validates the inhibitory activity of the original molecules and rules out the possibility of a false positive effect. Administering SPI-24 and SPI-77 to BACHD mutant mice using different methods, directly to the striatum, by oral delivery or subcutaneous injection, significantly reduced the expression level of mutant *Htt* mRNA in the striatum and increased the level of BDNF. These results, together with the pharmacokinetics experiments, provide strong evidence that both SPIs can pass the BBB and lower mutant *Htt* expression in the striatum. Importantly, lowering mutant *Htt* levels by SPIs led to a significant attenuation of motor and anxiety abnormalities that are associated with the HD mouse phenotype. Moreover, SPIs administration to early-stage BACHD mice slowed down the progressive deterioration of the disease.

Spt5, the target of SPIs, is an important ubiquitous transcription elongation factor. Hence, a major concern is that Spt5 inhibition

may affect many genes and lead to undesired side effects. However, several lines of evidence support the selectivity of SPIs described here. First, our previous study reported two sets of SPIs: general inhibitors that affect all known Spt5 functions and selective SPIs that primarily affect a single function (Bahat et al, 2019). The SPIs used in this study belong to the selective class as they predominantly affect Spt5's ability to promote the mutant *Htt* transcription, while their effect on its other activities is minor. In addition, the downregulation of Spt5 in mammalian cells has a limited effect on global gene expression (Diamant et al, 2012, 2016b; Fitz et al, 2018; Komori et al, 2009; Pavri et al, 2010; Rahl et al, 2010; Stanlie et al, 2012) most likely due to functional redundancy with other elongation factors. Thus, the impact of the selective SPIs is likely even more limited. Indeed, short- and long-term SPI-24 treatments had no global effect on mRNA levels in mice. Furthermore, no apparent toxicity was observed following prolonged treatment of mice with SPI-24, including food and water intake, body composition and multiple blood parameters, consistent with its selective effect. Nevertheless, as with all drugs, off-target effects cannot be ruled out.

HD is characterized by a progressive deterioration of motor functions. Our findings reveal the beneficial effect of SPIs on HD-associated motor phenotype. Specifically, SPI-24 significantly reduced the slip ratio in HD mice that typically display a strong motor phenotype at the age of 7–8 months. When given to early-stage HD mice, it successfully prevented the impairment of their motor coordination and balance (Table 1). This improvement was also reflected in their enhanced climbing capability and performance on the running wheel. In addition, their involuntary hyperactive movement was reduced following SPI-24 treatment. SPI-77 also had a positive effect on their motor functions; however, this effect was moderate relative to SPI-24, probably due to its lower stability during long-term treatments.

Anxiety is very common in HD (Dale and van Duijn, 2015; Hult Lundh et al, 2013) and we, therefore, assessed the effect of the SPIs on the mice's response to abrupt auditory stimuli (ASR test). Treated mice were habituated to the repeated startling stimuli similarly to healthy WT mice, indicating that both SPIs reduced their HD-associated anxiety. Unexpectedly, this improvement was not observed following long subcutaneous delivery of SPI-24 (Fig. EV5B). This might be the consequence of a decline in the stability and the effective concentration of the molecule over a long period in the pump as determined from the LC/MS measurements (Fig. EV5C). These findings suggest that continuous supplementation is necessary (Fig. EV4A) in order to maintain the relief of the anxiety symptoms (Fig. 4B), consistent with the study indicating

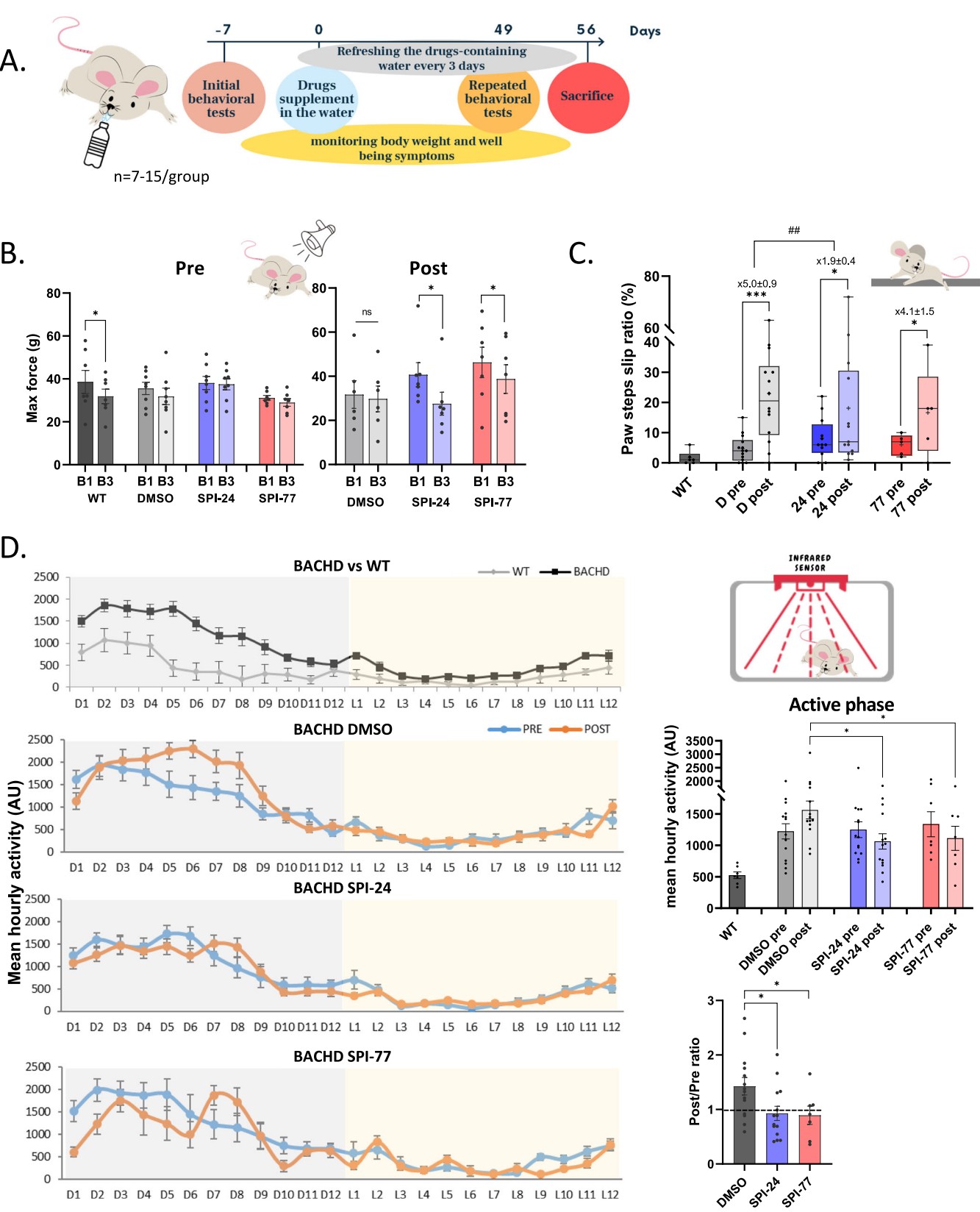

**Figure 4. The effect of long-term oral delivery of SPIs on HD disease symptoms.**

(A) The study design of long oral treatment of 3-month-old BACHD mice. (B) Acoustic startle response test. The response of untreated WT and BACHD mice to stimuli in B1 and B3 pre-treatment (left) and after 2 months of SPIs administration (post, right). Each bar represents the means ± SEMs of 7 mice. (C) Beam walk test. Paw slip ratio of untreated WT and BACHD mice pre vs post treatments. The lines represent the median of 6–14 mice. The ratio between post- and pre-treatment is calculated for each mice and the mean number ± SEM is presented. (D) Home cage locomotion assessments (48 h). Top left panel: WT vs BACHD mice before treatment. 2nd–4th left panels: BACHD mice pre vs post treatments with DMSO, SPI-24 and SPI-77, respectively. Dots represent means ± SEM of 7–15 mice. Top right panel: Active (dark) phase mean hourly activity of the data presented in the four left panels. Bottom right panel: The ratio between post- and pre-treatment calculated for each mice. Each bar represents the mean ± SEM of 7–15 mice. Data information: The lines and the '+' within the box-whisker plots (min to max) in panel (C) represent the median and the average, accordingly. Asterisks in panel (C) denote statistically significant differences relative to DMSO according to Student's paired one-tailed $t$-tests. The statistical significance of the difference between the mean ratios is calculated by Student's unpaired two-tailed $t$-tests. Asterisks in panel (D) denote statistically significant differences relative to DMSO according to Student's unpaired two-tailed $t$-tests. $*p < 0.05$; $***p < 0.005$; $\#\#p < 0.01$; ns, not significant. Source data are available online for this figure.

that the neural mechanisms of motor and anxiety phenotypes may differ (Gil and Rego, 2008). Future medicinal chemistry studies are needed to improve stability for long-term delivery.

HD is a gain-of-function disease, but several studies suggest that the reduction of wt HTT level may also contribute to the disease phenotype (Cattaneo et al, 2001, 2005). Therefore, strategies aiming to lower *Htt* gene are expected to affect the already low wt *Htt*. The great advantage of the SPIs is the selective lowering of mutant *Htt* but not the wild-type gene. Another major benefit of the SPIs is their small size and chemical properties, enabling them to penetrate through the BBB and reach their target in the brain, which makes their delivery methods simple and easy.

The HTT protein is widely expressed throughout the body (Hoogeveen et al, 1993; Li et al, 1993; Trottier et al, 1995) and is known to be involved in several cellular processes (Cattaneo et al, 2005). Therefore it is not surprising that HD patients suffer from several peripheral symptoms in addition to the well-characterized neuropathology. These symptoms include changes in the muscles, circulation, and digestive systems, weight loss and increased pro-inflammatory signaling (Sassone et al, 2009; van der Burg et al, 2009; Wild et al, 2011; Björkqvist et al, 2008). Therefore, systemic treatment with SPIs might substantially improve the quality of life of patients with HD, in addition to the *Htt* lowering in the brain.

In summary, we have identified and characterized two small molecules that effectively relieve HD-associated phenotypes and slow down their progression in preclinical studies. Their favorable effects are based on the selective lowering of the mutant *Htt* and their chemical properties that enable them to pass the BBB. These SPIs provide a great therapeutic strategy due to their simple delivery, especially when prolonged and safe treatment of HD patients is needed. Thus, SPIs are excellent candidates for clinical evaluations as a therapy against HD.

# Methods

## Cells and SPIs treatment

STHdh Q111 and STHdh Q7 striatal cell lines (Cat# CH00095 and CH00097, respectively; Coriell Institute for Medical Research) were grown in Dulbecco's modified Eagle's medium supplemented with 10% fetal calf serum, 1% penicillin-streptomycin and 0.4 mg/ml G418 at 33 °C. The cells were re-plated no more than 5–6 times, as recommended. Cells were incubated with SPIs either for 48 h to determine the effect on steady-state mRNA levels or for 2 h in the presence of 4-thiouridine to analyze newly synthesized RNA. The

HD patient-derived cells were also obtained from Coriell Institute (HD 19/44Q: GM03643; HD 15/55Q: GM13507; HD 16/66Q: GM13515, HD 18/180Q: GM09197; healthy control: ND03231) and they were cultured under standard conditions (37 °C and 5% $CO_2$) either with RPMI 1640 media supplemented with 1% Penicillin-Streptomycin and 15% fetal bovine serum (Lymphoblast cells), or using MEM-Eagle with Non-Essential Amino Acids (MEM-NEAA; #01-040-1A), supplemented with 1% Penicillin-Streptomycin, 1% Glutamax (#35050-038, Thermo Scientific) and 15% fetal bovine serum (fibroblast cells).

## Compounds and medicinal chemistry

SPI-24 (Z54716045) and SPI-77 (Z285974290) were purchased from Enamine. All other analogs were procured as listed in Appendix Table S4. Chemical synthesis, structure-activity relationship (SAR) by-catalog, quality control tests of the compounds and part of the stability tests of the compounds were performed by the Medicinal Chemistry unit of the Nancy and Stephen Grand Israel National Center for Personalized Medicine (G-INCPM, Weizmann Institute of Science).

SPI-4516 synthesis: all reagents and solvents used for the synthesis were purchased from Sigma-Aldrich, Merck and Acros. Chemical building blocks were purchased from Enamine and MolPort chemical suppliers. Flash chromatography was performed using atomized CombiFlash® Systems (Teledyne Isco, USA) with RediSep Rf Normal-phase Flash Columns. Reaction progress was monitored by Waters UPLC-MS system: Acquity UPLC® H class with PDA detector, and using Acquity UPLC® BEH C18 1.7 µm 2.1 × 50 mm Column (PN:186002350, SN 02703533825836). MS-system: Waters, SQ detector 2. 1H and 13 C NMR spectra were recorded on a Bruker Avance - 500 MHz spectrometer, equipped with QNP probe. Chemical shifts are reported in ppm on the δ scale down field from TMS. All J values are given in Hertz. In a micro wave vial were added (Appendix Fig. S1): (1) 4-bromo-N-(2-ethylphenyl)benzenesulfonamide (58 mg, 0.17 mmol; MolPort-002-826-063), (2) pyridazin-3(2H)-one (25 mg, 0.26 mmol), $K_2CO_3$ (36.6 mg, 0.27 mmol), quinolin-8-ol (2.4 m, 17 µmol) CuCl (1.7 mg, 17 µmol) and dissolved in DMF (2 ml). Then the vial was capped with a crimp cap flushed 3 X with argon and heated in the micro wave reactor (Biotage Initiator) at 140 °C for 14 h. The reaction mixture was then poured into water (10 ml) and extracted 3 X EtOAc. The combined organic layers where washed with water and brine then dried on $Na_2SO_4$. The reaction mixture was then concentrated under vacuum and chromatographed, DCM to EtOAc 16 min gradient. N-(2-ethylphenyl)-4-(6-oxopyrida-zin-1(6H)-yl)benzenesulfonamide (3) eluted in 50% EtOAc which was

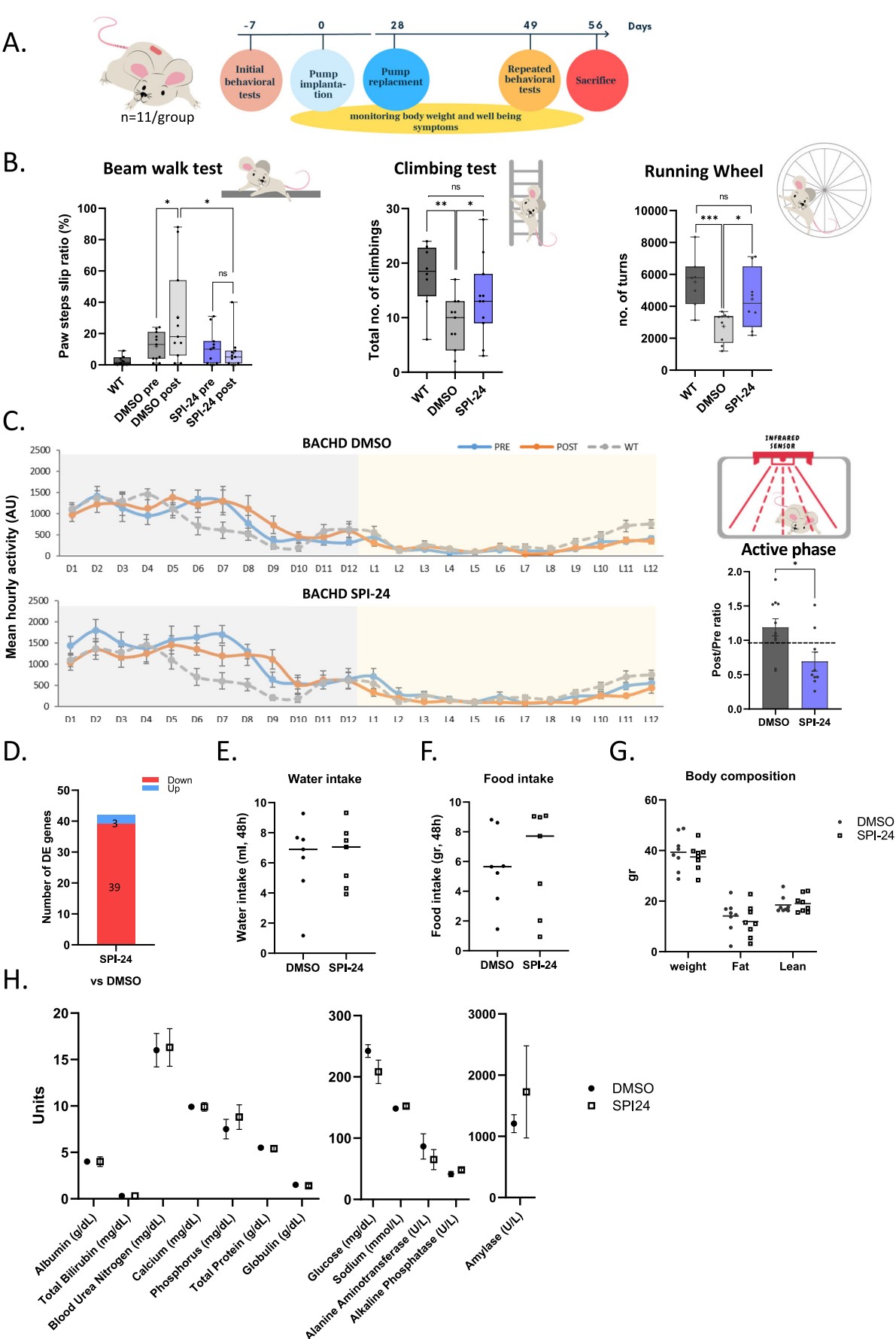

**Figure 5. Analyses of HD disease symptoms following long-term subcutaneous delivery of SPI-24.**

(A) The study design of long subcutaneous treatment of 3 months old BACHD mice. (B) Motor functions tests performed to assess the effect of long treatment of SPI-24 on HD progression. Left: Beam walk test of WT and BACHD mice pre- and post-treatment; middle: Climbing test of untreated WT and treated-BACHD mice; right: Wheel test of untreated WT and treated-BACHD mice. Each line represents the median of 7–11 mice. (C) Home cage locomotion assessment (48 h) of WT and BACHD mice pre and post treatment. Dots represent means ± SEM of 8–11 mice. The right panel shows the ratio between post- and pre-treatment calculated for each mice in the active (dark) phase. Each bar represents the mean ± SEM of 8–11 mice. (D) The global effect of long-term SPI-24 treatment on the number of differentially expressed genes, compared to DMSO ($n = 5$). Sequencing libraries from mRNA extracted from the striatum of treated mice were prepared and subjected to deep sequencing. Reads were aligned to the mouse genome. (E,F) Water and food intake of 8 weeks SPI-24 treated mice relative to DMSO, during 48 h. Each line represents the median of 7 mice. (G) The body composition of the treated mice relative to DMSO. Each line represents the median of 8 mice. (H) Chemical analysis of blood samples taken from the 8 weeks treated mice relative to DMSO. Each dot represents means ± SEM of 4 mice. Data information: The lines and the '+' within the box-whisker plots (min to max) in panel (B) represent the median and the average, accordingly. Asterisks in panel (B) denote statistically significant differences according to Student's unpaired two-tailed (middle and right panels) or paired one-tailed (left panel) t-tests. Asterisks in panel (C) denote statistically significant differences according to Student's unpaired two-tailed. *$p < 0.05$; **$p < 0.01$; ***$p < 0.005$; ns, not significant. Source data are available online for this figure.

concentrated to give a white solid. MS (ES): $m/z$ calc ([M + H] = 356.1); found (356.3). 1 H-NMR (500 MHz, (CD3)2SO): 9.71 (s, 1H), 8.11 (dd, J = 5, 2 Hz, 1H), 7.82 – 7.78 (m,4H), 7.52 (dd, J = 9, 4 Hz, 1H), 7.24 – 7.19 (m, 1H), 7.18 – 7.15 (m, 1H), 7.12 − 7.07 (m, 2H), 6.91 (dd, J = 9, 2 Hz, 1H), 2.52 − 2.49 (m,2H), 0.98 (t, J = 8 Hz, 3H).

The long-term stability tests of SPI-24 and SPI-77 in drinking water containing 25% fructose were carried out by Bienta (Enamine Biology Services). 62.5 μl of SPI-24 (20 mM in 100% DMSO) and SPI-77 (2 mM in 100% DMSO) were mixed with 4.4 ml of drinking water containing 25% previously dissolved fructose. After preparation, all formulations were kept at room temperature for one week. Five-time points over one week were analyzed by HPLC-MS: 0, 1, 3, 5, and 7 days (in duplicates; Table EV2A). Reactions were stopped by adding cold methanol with internal standards Fluazifop-p-butyl (for SPI-24; dilution 1:600) and Fludioxonil (for SPI-77; dilution 1:30). Samples prior to HPLC/MS analysis were stored at −25 °C in the polypropylene tubes.

Screening of preclinical formulations to obtain the maximal consistent concentration of SPI-24 and SPI-77 was carried out by Bienta. The compounds were assessed for solubilization concentrations of ≥10 mg/ml over dilution steps using ten formulation vehicles/solvents known to be compatible with ALZET pumps. The extent of the compound solubilization was evaluated visually. The outcome of this analysis is presented in Table EV2B.

## Animals

The HD mice model used in this study is the BACHD (FVB/N-Tg(HTT*97Q)IXwy/J; Stock No: 008197 from Jackson labs). BACHD mice express a neuropathogenic, full-length human mutant Huntingtin gene containing 97 mixed CAA-CAG repeats encoding a continuous polyglutamine (polyQ) stretch (Gray et al, 2008). The colony was maintained by crossing WT FVB/N females (Strain #001800, Jackson labs) with mutant heterozygous BACHD males in our breeding facility to obtain male and female offspring, either WT or heterozygous for the BACHD transgene. The offspring's genotype was determined by PCR of their genomic DNA as recommended by the Jackson laboratory. After weaning, littermates were housed (wild-type and HD mice together, maximum 5 in each cage) under a 12/12 h light/dark cycle and were fed ad libitum. Mutant mice were randomly divided between treatment groups and each group contained similar numbers of females and males of similar age and weight (see Appendix Table S1 for mice list, ages and treatments). WT mice from the same littermates were left untreated. All procedures employed on

experimental animals, including their transportation, routine care, and use in experiments, were conducted in accordance with the Israel animal welfare law and guidelines and approved by the Weizmann Institute council for Experiments on Animals (IACUC protocols numbers: 19191219-15, 01660220-3, 06450820-2, 00900122-2). During all the experiments described below, mice were monitored for their behavior, activity, CNS signs, stress or pain signs, and weight.

## Pumps implantation for direct brain or subcutaneous injections

For the direct brain injection, the right hemisphere of BACHD mice was implanted with a 28-gauge infusion cannula (brain infusion kit 2; Alzet), connected by a small vinyl catheter tube (3 cm long) to a subcutaneous osmotic minipump (model 1004; Alzet) that constantly infused the SPIs for 28 days in a constant low rate (0.11 μl/h) (Fig. 2A). To ensure their proper operation, the pumps were pre-filled with 100 μl SPIs (SPI-24: 5 μg/day; SPI-77: 0.6 μg/day) or DMSO vehicle (25% in saline solution) 48 h prior to implantation and incubated in a sterile saline solution at 37 °C, according to the manufacturer's instructions. Figure EV2A, depicts the stereotaxic position of the implanted cannula's tip in the right striatum; coordinates (with respect to the Bregma): AP (Antero-Posterior): +0.62 mm; ML (Medium-Lateral): +1.75 mm; DV (Dorso-ventral): −4 mm; based on the stereotaxic mouse brain atlas of Paxinos and Franklin (George Paxinos and Keith Franklin, 2019) and Perucho et al (Perucho et al, 2013). The cannula was mounted to the skull using carboxylate cement (3 M™ Durelon™).

For the long-term subcutaneous pump treatment (8 weeks) we used the 2004 model Alzet osmotic minipump that constantly infused the SPIs for 28 days (pumping rate of 0.25 μl/h). The pumps were pre-filled with 200 μl DMSO-PEG400 (50%:50%) or SPI-24 at a high concentration (187 μg/day) 48 h prior to implantation and incubated in sterile saline solution at 37 °C. After 28 days, the pump was replaced with a new one.

Pain management: In all surgeries, mice were anesthetized with isoflurane (4% induction, ~2% maintenance), and Buprenorphine (0.1 mg/kg) was injected SC to prevent pain development. In addition, Lidocaine (5 mg/kg) was injected into the site of the scalp incision. Post-surgery and during the first 3 days after, the mice were injected SC with Carprofen (5 mg/kg). At the endpoint of the experiments, the mice were sacrificed using $CO_2$ and their brain was quickly removed and placed in ice-cold saline.

## Oral administration

Short-term treatment: DMSO (25%) or SPIs (490 µg/day SPI-24 or 55 µg/day SPI-77) were diluted in drinking water (total volume: 250 µl) and directly administrated into the mice's stomach using an oral gavage needle (22 G, 3.8 cm long) for 4 consecutive days. Afterward, the mice were sacrificed, and their brains were removed for further analysis.

Long-term treatment: DMSO or SPIs (SPI-24: 35 mg daily; SPI-77: 4 mg daily) were provided to the mice in their drinking water (containing 8% fructose which was found to be the minimum concentration that made the SPIs-containing water tolerable for drinking) for 2 months in light-protected bottles. The volume of the regular daily-consumed water was previously determined to be ~6.5 ml, and the daily SPIs concentrations were calculated based on this observation. Water-containing DMSO/SPIs were refreshed every 3 days. We measured the residual volume of the DMSO/SPIs drinking water and found no difference in the leftover volumes between DMSO and the SPIs.

## Behavioral and diagnostic tests

All behavioral tests were performed during the dark phase following at least 1 h of habituation to the test room. All tests and analyses were carried out blinded to the mice genotype and treatment groups. No significant differences were observed between sexes.

### Acoustic startle response (ASR) test

Mice's sensorimotor functions and anxiety, measured as a force response to repeated stimuli, were assessed in a Startle Response System (TSE Systems). The test protocol was adapted from Lebow et al (Lebow et al, 2012). Briefly, the experiment comprised three blocks (B1–3). In the first and third blocks, a stimulus of 120 db was presented 6 times (B1, B3 stimulus). At B2, a set of 12 presentations of 120 db, or 74 db, 78 db, and 82 db stimuli linked to a 120 db stimulus was generated randomly (this served as pre-pulse inhibition). As a control, the system measured the response of the mice to 'non-stimulus' of 65 db.

### Beam walk test

The beam walk test is utilized to assess motor coordination and balance and was adapted from Appel et al (Appel et al, 2016). Following a short training session, mice were forced to walk, 5 times without any stops or turns, on a narrow beam (50 cm long, 5 or 10 mm wide), leading them to their empty home cage. These test sessions were video recorded using an overhead camera and later were manually scored and analyzed for the following parameters: the time of crossing the beam (sec), the number of paw steps, and the falls ratio (the percentage of the number of falls relative to the number of steps).

### Home-cage locomotion (HCL) test

BACHD and WT littermates were housed individually in cages with an infrared sensor (InfraMot; TSE Systems) to monitor their activity over 72 h with a 12/12 h dark/light cycle. The first 24 h were discarded from the final analyses to exclude bias due to acclimation to the new environment. The mean hourly activity during the dark/active and light/inactive phases of the mice was calculated.

### Climbing test

Mice were placed in an open metal wire cylinder (diameter: 10.5 cm, height: 15.5 cm, adapted from Hickey et al (Hickey et al, 2005)) for 5 min, and their climbing behavior was video-recorded for later analysis. Climbing was defined as a state when all four paws of the mouse were off the floor of the testing bench and on the wall of the climbing cage. The total number of climbing was then counted.

### Metabolic cages

Metabolic measurements, including food and water intake, were measured for three consecutive days using the PhenoMaster system (TSE-Systems, Berlin, Germany). It consists of a combination of sensitive and precise weighing sensors that automatically measure the amount of food and liquid consumed over time. The metabolic cages are located within a climate cabinet with a controlled temperature of 22 °C, 40% humidity, and the same light/dark cycle as in the housing room.

### Voluntary wheel running

Mice were individually placed in metabolic cages equipped with a vertical running wheel (11.5 cm diameter, TSE Systems, Berlin, Germany), in order to assess their voluntary physical activity. Data from the first dark phase of the wheel-running activity were automatically collected and processed with PhenoMaster software (TSE Systems) to measure the total number of rotations (right and left).

### Body composition

Mice were held in a restraint cylinder in order to inhibit movement. The cylinder was inserted to a MiniSpec LF50 body composition analyzer (Bruker) for approximately 1 minute for body composition measurements. The analyzer uses nuclear magnetic resonance to quantify lean and fat masses.

## Blood chemistry diagnosis

The levels of different biochemical markers in the mice's plasma (100 µl) were determined using a chemistry analyzer, VetScan VS2 (Abaxis North America), using VetScan Comprehensive Diagnostic Profile reagent rotor (#500-038).

## RNA and protein analysis

Upon removal of the brain, it was rinsed in ice-cold saline and placed in a pre-chilled brain matrix (Mouse/40–75 g/Coronal/1 mm; catalog no. 68707 from RWD). The brain was sliced using a single-edge blade of carbon steel (BN71960, Bar Naor, Israel) to 2 mm slices and placed on dry ice. Then small punch-size pieces of tissue (~10 mg) from the striatum were removed for RNA and protein analysis (Fig. EV2B). Tissues for protein extraction were immediately frozen at −80 °C till use. Tissues for RNA extraction were immediately placed in Bio-Tri RNA (Bio-Lab Chemicals), vortexed well, and sonicated (10 cycles for 30 s on/off each cycle). The extracts were then cleared by centrifugation at $15,000 \times g$ for 10 min, and total RNA was further extracted using Direct-zol™ RNA MiniPrep kit (Zymo Research). cDNA was synthesized from 200 ng of total RNA using the High-Capacity cDNA Reverse Transcription Kit (ABI, Thermo Fisher Scientific) and random

hexamers. cDNA samples were analyzed by quantitative PCR (qPCR) in a QuantStudio™ 6 Flex Real-Time PCR System using Fast qPCRBIO SYBR Green Mix (PCR Biosystems). The primer sequences are shown in Appendix Table S5.

For protein extraction, tissues from the striatum were sonicated in RIPA buffer (150 mM Sodium chloride, 50 mM Tris-HCl pH 8.0, 1% Nonidet P-40, 0.5% Sodium deoxycholate, 0.1% SDS) supplemented with 6 M UREA and protease inhibitor cocktail (1:100; K1007, ApexBio), and cleared by centrifugation at $15,000 \times g$ for 10 min. Total protein concentration was determined using Pierce BCA protein assay kit (Thermo Fisher Scientific). 25–40 µg of protein were resolved using 6% SDS-PAGE gel and blotted onto a nitrocellulose membrane. Samples were probed with anti-huntingtin EP867Y (1:1000, ab45169; Abcam), anti-PolyQ clone MW1 (1:1000; MABN2427, Merck) and anti-eIF4G1 for human samples (1:2000; 07-1800, Merck) or anti-eIF4G3 for mouse samples (1:5000; GTX118109, GeneTex) as normalized proteins. Western blots were quantitated using Image Studio™ Software.

## DNA extraction from the plasma

Blood was collected using the submandibular vein method with 21 G needle directly into MiniCollect® Lithium Heparin tubes (Cat # 450537; Greiner). Plasma was obtained by 10 min centrifugation at 2000 rpm. 25 µl were taken for DNA extraction using the Quick-DNA™ MiniPrep kit (Cat #D3024; Zymo Research). The level of the mitochondrial ND2 in the extracted DNA was analyzed by qPCR and was normalized to beta-actin. The primer sequences are shown in Appendix Table S5.

## Pharmacokinetics

To test the stability of SPIs in the mice plasma over time as well as their penetration ability through the blood-brain-barrier (BBB), each SPI was subcutaneously injected into WT mice at two different quantities [SPI-24: 38 and 190 µg (2-10 mg/Kg); SPI-77: 4.4 and 22 µg (0.2–1.1 mg/Kg)], 12 mice for each compound quantity. Blood samples were taken at 8 times point after injection (twice from each mouse) and centrifuged at 2000 rpm for 10 minutes to obtain plasma. After 12 h the mice were sacrificed, and their brain was removed for analysis. Three mice were left untreated (control), bled at zero time and sacrificed for brain removal (the total number of mice in this experiment was 51). This experiment was carried out by Science in Action, a preclinical research company (Https://www.sia10.com/). The samples were snap freeze at -80°C and transferred to the Targeted metabolomics unit at the Weizmann Institute of Science. Brain and plasma samples were dried in a freeze-dryer. Then, 800 µl acetone was added, the samples were homogenized using a handle pestle mixer and then shaken (Thermomixer, 10 °C, 10 min). Following centrifugation (21,000 X g, 10 min), the supernatants were collected, and the pellet extraction was repeated with 800 µl methanol. The acetone and methanol supernatants were combined, and SPI-8690 (SPI-24 analog; CAS 312271-66-2) was added as an internal standard. Samples were dried in SpeedVac and freeze dryer. Then, the dry residues were re-suspended in 100 µl of methanol with short vortex and sonication, centrifuged (21,000 × g, 5 min), and transferred to nano-filter vials (0.2-µm PTFE, Thomson) for LC-MS/MS analysis. Quantification of SPI-24 was carried out using an Acquity I-class UPLC system coupled to Xevo TQ-S triple quadrupole mass spectrometer (both Waters, US).

The UPLC was performed using a BEH C18 column (2.1 × 100 mm, 1.7 µm; Waters). Mobile phase A was 5% aqueous acetonitrile, and mobile phase B - acetonitrile, both with 0.1% formic acid. The flow rate was kept at 300 ml/min with a linear gradient of B from 50 to 95% for 4.5 min. The column temperature was set at 35 °C, and the injection volume was 3 µl. An electrospray ionization interface was used as an ionization source. Analysis was performed in positive ionization mode. The compounds were detected using multiple-reaction monitoring, using argon as the collision gas: for SPI-24, $391 > 73$ and $391 > 206.1$ m/z, with collision energies 17 and 11 eV, respectively, and for SPI-8690, $325.1 > 105.1$ and $325.1 > 158$ m/z, with collision energies 237 and 13 eV respectively. TargetLynx (Waters) was used for data analysis. The same procedure was performed for brain samples following short- and long-term SPIs treatments.

## High-throughput sequencing of total mRNA (RNA-Seq)

The RNA-seq libraries were prepared with the TruSeq Stranded mRNA Library Prep (Cat #20020594; Illumina). They were multiplexed and sequenced at the iGE3 Genomic Platform of the University of Geneva on a HiSeq 4000 Illumina sequencer, and at the Genomic unit of the G-INCPM (Weizmann Institute of Science) on a NextSeq 500 High Output as single reads (read length 50 base pair). The RNA-seq analysis of the short SC treatment yielded 25 M reads per sample on average ($n = 3$–5), and the prolonged treatment yielded 40 M reads per sample on average ($n = 5$ for each group). The bioinformatics analysis was carried out as follows: Poly-A/T stretches and Illumina adapters were trimmed from the reads using cutadapt. Reads were mapped to the M. musculus reference genome GRCm38 using STAR, supplied with gene annotations downloaded from Ensembl. Reads shorter than 30 bp and reads that mapped to more than one region were discarded. Expression levels for each gene were quantified using htseq-count and differentially expressed genes (Fold Change ≥ 2) were identified using DESeq2. Raw P values were adjusted for multiple testing using the procedure of Benjamini and Hochberg. Pipeline was run using snakemake.

## Statistical analysis

The data are presented as median and mean ± SEM of biological replicates. For the statistical analysis, mean values were used. For two unpaired groups (i.e. DMSO vs treatment or WT vs mutant), a two-tailed Student's $t$-test was used. For the comparison of two paired groups (pre vs post) one-tailed $t$-test was used. All datasets met the assumptions for parametric t-tests according to 4 tests: Anderson-Darling (A2*), D'Agostino-Pearson omnibus (K2), Shapiro-Wilk (W), and Kolmogorov-Smirnov (distance). In few cases, where the populations had different standard deviations, the $t$-test was corrected with Welch $t$-test. Significance symbols in all experiments are: *$p < 0.05$; **$p < 0.01$; ***$p < 0.005$; ****$p < 0.001$ (calculated by GraphPad Prism).

# Data availability

The RNA-seq datasets generated during this study have been deposited in NCBI's Gene Expression Omnibus (Edgar et al, 2002) and are accessible through GEO Series accession number GSE225790.

### The paper explained

#### Problem

Huntington's disease (HD) is a neurodegenerative disease caused by an abnormal expansion of CAG repeats in the huntingtin (*Htt*) gene. Mutant *Htt* is prone to aggregation and leads to neuronal degeneration and profound motor and psychological abnormalities. The most promising therapeutic strategy for HD aims to reduce mutant *Htt* expression, while sparing unmutated *Htt*. The transcription elongation complex Spt4/Spt5 is specifically required for mutant *Htt* transcription. We previously identified the first Spt5 small molecule inhibitors called SPIs. Few SPIs selectively inhibit the expression of the mutant *Htt* gene without affecting the expression of the normal gene or other Spt5 activities. While these results validated Spt5 as a potential drug target for HD, it remains unclear whether SPIs can be improved or serve as effective therapeutics in animal models.

#### Results

We identified two more potent SPI inhibitors (SPI-24 and SPI-77) that selectively reduce mutant but not wild-type *Htt* expression in HD cells at low concentrations. Pharmacokinetic studies revealed that these potent SPIs pass the blood-brain barrier. In the BACHD mouse model of HD, direct brain delivery, oral administration, and subcutaneous injection of these SPIs preferentially lower mutant *Htt* mRNA and protein levels in the striatum. In addition, SPIs restored HD biomarkers, improved motor and anxious-like phenotypes and delayed disease progression. SPI-24 long-term treatment had no side effects or global changes in gene expression. Thus, lowering mutant Htt levels by small molecules can be an effective therapeutic strategy for HD.

#### Impact

Our study demonstrated that lowering mutant Htt levels by small molecules targeting Spt5-Pol II represents a highly effective therapeutic approach for HD. Due to their simple delivery, the SPIs are excellent candidates for further clinical evaluations as a potential therapy against HD.

# Peer review information

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

## Acknowledgements

We would like to thank Dr. Efrat Ben-Zeev from the Medicinal Chemistry unit of the Nancy and Stephen of G-INCPM (Weizmann Institute of Science) for performing the SAR-by-catalog search; The Crown Genomics Institute of the Nancy and Stephen G-INCPM and the iGE3 Genomic Platform of the University of Geneva for RNA-Seq services; Dr. Eviatar Weizman from the Mantoux Institute for Bioinformatics of the Nancy and Stephen G-INCPM for the RNA-Seq data analysis; Dr. Shelly Zinamon for her help and guidance with animal care and treatment; Dr. Yuri Kuznetsov for his help with animal perfusion technique; Science in Action company for performing the PK experiment; Bienta department in Enamine for the formulations screen service. This work was supported by external grants from The Israel Innovation Authority Kamin and from the Hereditary Disease Foundation and by Weizmann Institute internal grants from Dr. Barry Sherman Institute for Medicinal Chemistry; David and Fela Shapell Family Center for Genetic Disorders Research; Weizmann - Center for Research on Neurodegeneration; the Estate of Manfred and Margaret Tannen and Joel and Mady Dukler Fund for Cancer Research and a fellowship from the Society of Swiss Friends of the Weizmann Institute of Science to BW. MT is the incumbent of the Carolito Stiftung Research Fellow Chair in Neurodegenerative Diseases; YK is incumbent of the Sarah and Rolando Uziel Research Associate Chair; RD is the incumbent of the Ruth and Leonard Simon Chair of Cancer Research.

## Author contributions

**Anat Bahat**: Conceptualization; Data curation; Formal analysis; Funding acquisition; Investigation; Methodology; Writing—original draft; Project administration; Writing—review and editing. **Elad Itzhaki**: Investigation. **Benjamin Weiss**: Investigation. **Michael Tolmasov**: Methodology. **Michael Tsoory**: Investigation; Methodology. **Yael Kuperman**: Methodology. **Alexander Brandis**: Methodology. **Khriesto A Shurrush**: Methodology. **Rivka Dikstein**: Conceptualization; Data curation; Formal analysis; Funding acquisition; Investigation; Writing—original draft; Project administration; Writing—review and editing.

## Disclosure and competing interests statement

RD and AB declare a patent for the use of Spt5 inhibitors (U.S. Patent No. 20210379060). The rest of the authors declare no competing interests.

# Expanded View Figures

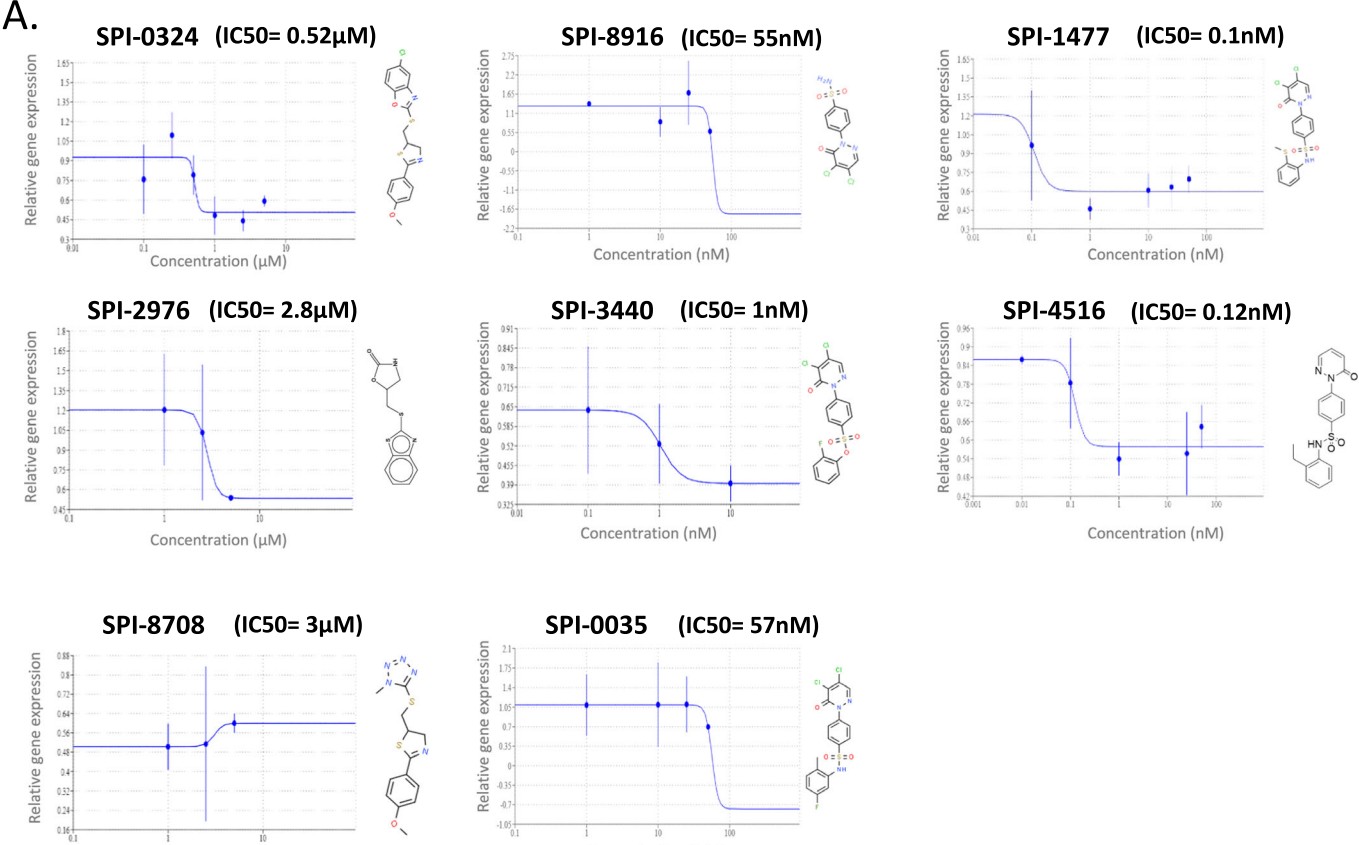

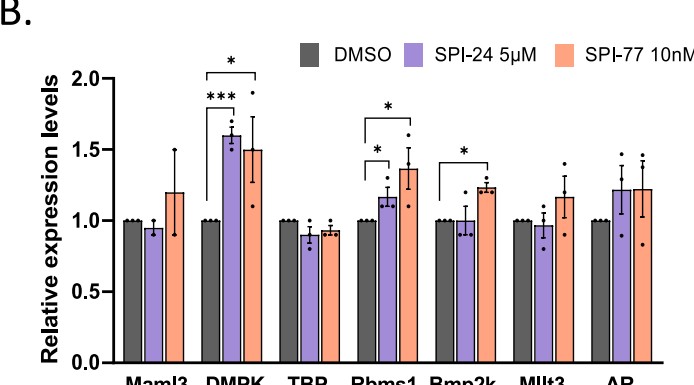

**Figure EV1. Characteristics of the different analogs.**

(A) The chemical structure and the IC$_{50}$ of the SPIs analogs that were found to be biologically active. (B) Q111 cells were metabolically labeled with 5-thiouridine for 2 h in the presence of DMSO, SPI-24 (5 µM) or SPI-77 (10 nM). Newly synthesized RNAs were purified, and mRNA levels of different genes were determined by qRT-PCR. Each bar represents the means ± SEMs of 3 independent experiments. Data information: The asterisks in panel (B) denote statistical significance differences relative to DMSO according to Student's unpaired $t$-test (one tailed). *$p < 0.05$; ***$p < 0.005$. Source data are available online for this figure.

A.

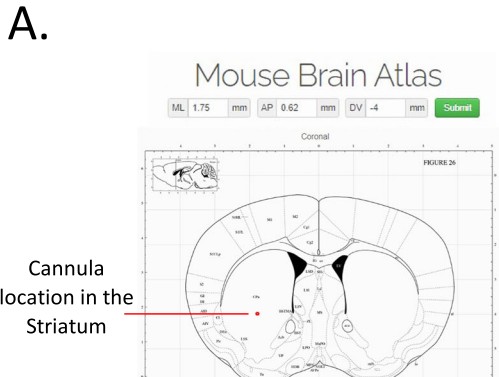

Cannula location in the Striatum

B.

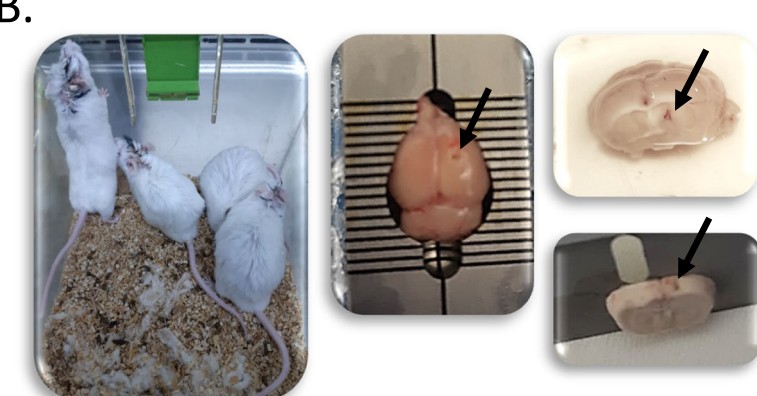

C.

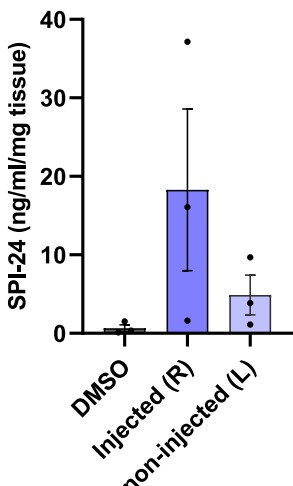

**Figure EV2. Direct injection of SPIs into the striatum.**

(A) Cannula implantation coordinates aiming the center of the right striatum (according to Perucho et al, 2013). (B) Left: post-operated mice. Middle: whole brain from treated mouse located in the brain matrix. Right: sliced brain from a treated mouse. The arrows indicate the injury place of the cannula. (C) The remaining concentration of SPI-24 in the mice's brain after 28 days of direct injection as was determined using LC-MS/MS and calculated as ng/ml and normalized to tissue weight. Data information: Each bar in panel (C) represents the means ± SEMs of 3 animals. Source data are available online for this figure.

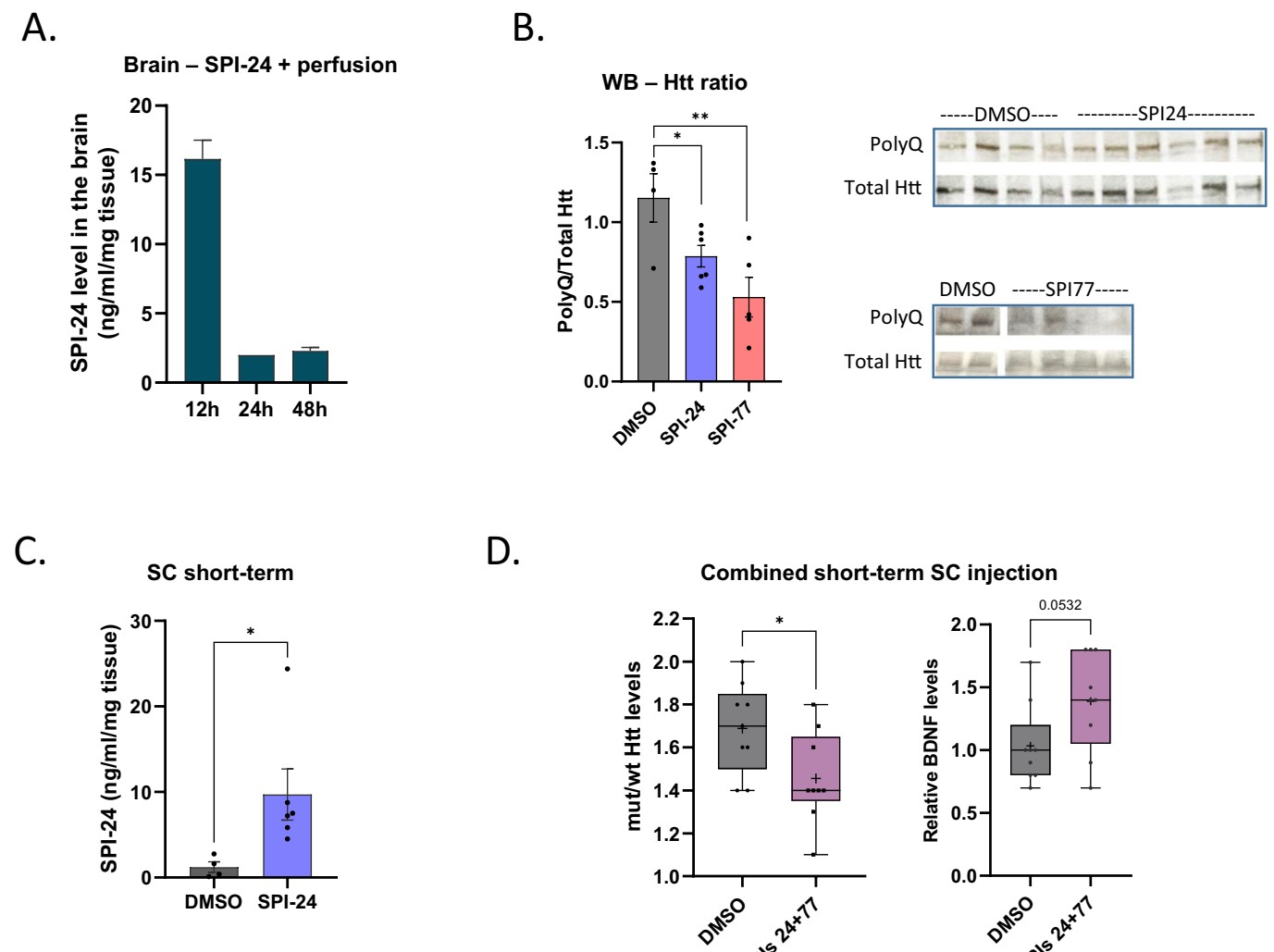

**Figure EV3.  Short-term subcutaneous injection of SPIs.**

(A) The concentration of SPI-24 in the mice's brain after 12, 24 and 48 h from the injection, measured by LC-MS/MS. Cardiac perfusion was performed in anesthetized mice before brain removal. Each bar represents the means ± SD of 2 animals. (B) Western blot analysis of the striatum of treated mice upon short-term subcutaneous injection showing the ratio of mutant (polyQ) vs total HTT proteins. Each bar represents the means ± SEMs of 4–6 animals. Representative blots are shown on the right. (C) The remaining concentration of SPI-24 in the mice's brain after 4 days of subcutaneous injection as was determined using LC-MS/MS and calculated as ng/ml and normalized to tissue weight. Each bar represents the means ± SEMs of 4–6 animals. (D) Combined short-term subcutaneous injection of the SPIs. Left: The ratio of the levels of mutant and wt *Htt* mRNA in the striatum of treated mice upon combined subcutaneous injection of SPI-24 (5 mM), and SPI-77 (0.5 mM) for four subsequent days. Right: The relative levels of BDNF in the striatum of treated BACHD mice. Each line represents the median of 9 mice. Data information: The lines and the '+' within the box-whisker plots (min to max) in panel (D) represent the median and the average, accordingly. The asterisks in panel (B–D) denote statistical significance differences relative to DMSO according to Student's unpaired two-tailed *t*-test. *$p < 0.05$; **$p < 0.01$. Source data are available online for this figure.

## A.

**Oral long-term**

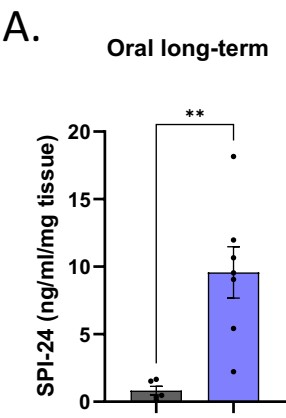

## B.

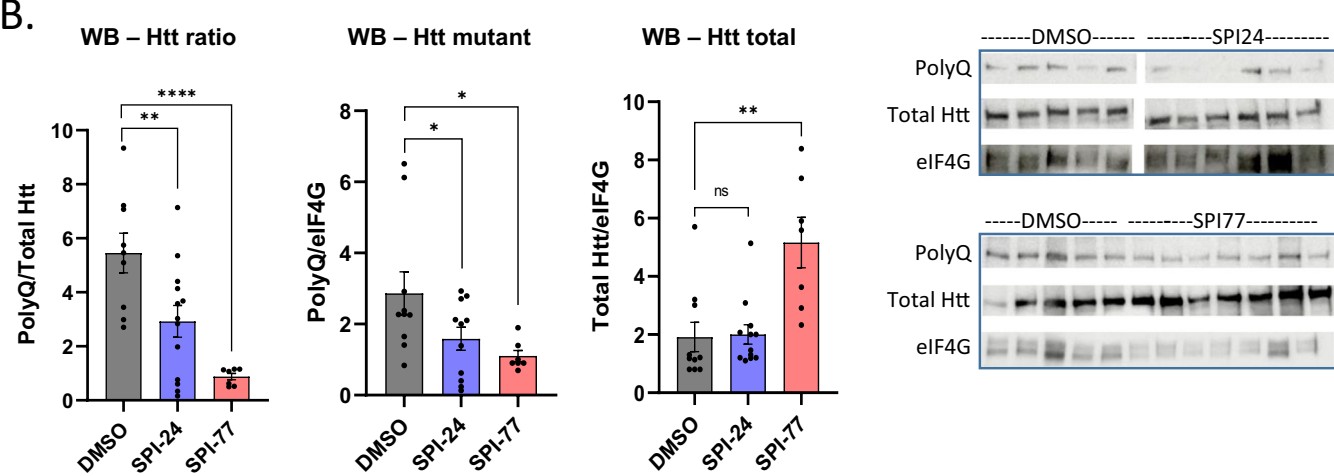

**Figure EV4.  Long-term oral delivery of SPIs.**

(**A**) The remaining concentration of SPI-24 in the mice's brain following long-term oral administration as was determined using LC-MS/MS and calculated as ng/ml and normalized to tissue weight. Each bar represents the means ± SEMs of 5–7 animals. (**B**) Western blot analysis of the striatum of treated mice upon long-term oral administration showing the level of mutant (polyQ) and total HTT proteins relative to normalized protein (eIF4G3). Each bar represents the means ± SEMs of 7–13 animals. Representative blots are shown on the right. Data information: The asterisks in panels (**A**) and (**B**) denote statistical significance differences relative to DMSO according to Student's unpaired *t*-test (one tailed). *$p < 0.05$; **$p < 0.01$; ****$p < 0.001$; ns, not significant. Source data are available online for this figure.

A.

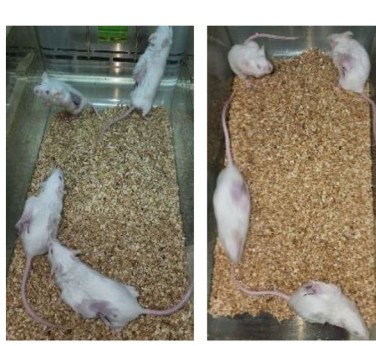

B.

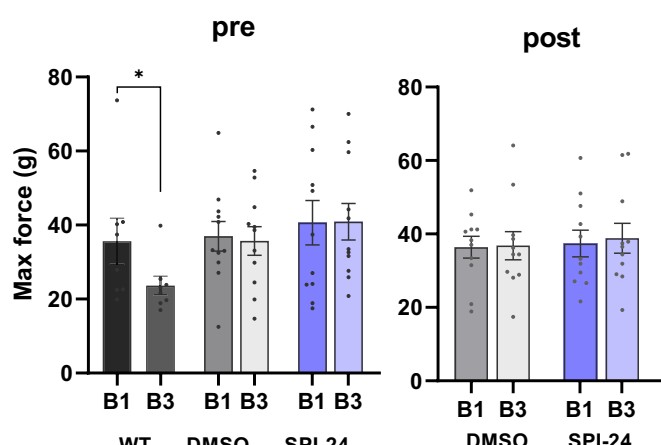

C. SC long-term

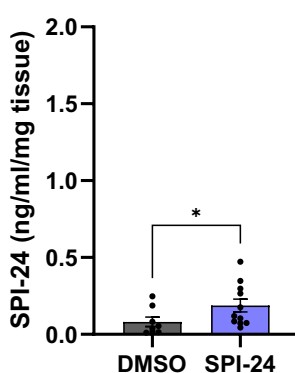

D.

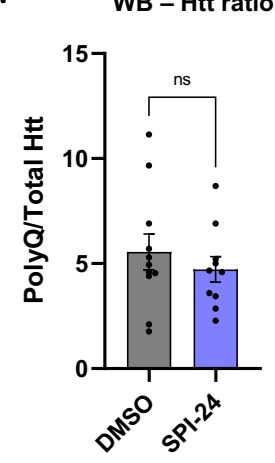

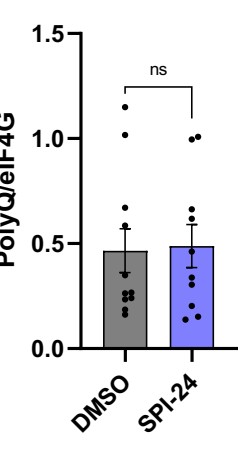

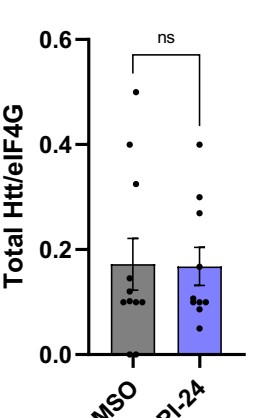

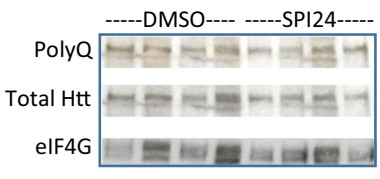

**Figure EV5. Long-term subcutaneous delivery of SPI-24.**

(A) Pictures of representative mice after pump implantation surgery. (B) Acoustic startle response test. The response of WT vs BACHD mice to stimuli in B1 and B3 pre-treatment (left) and after 2 months of subcutaneous administration of SPI-24 (right). Each bar represents the means ± SEMs of 11 mice. (C) The remaining concentration of SPI-24 in the mice's brain following long-term subcutaneous delivery as was determined using LC-MS/MS and calculated as ng/ml and normalized to tissue weight. Each bar represents the means ± SEMs of 9–11 animals. (D) Western blot analysis of the striatum of treated mice upon long-term subcutaneous delivery showing the level of mutant (polyQ) and total HTT proteins relative to normalized protein (eIF4G3). Each bar represents the means ± SEMs of 11 animals. Representative blots are shown on the right. Data information: The asterisks in panels (B) and (C) denote statistical significance differences relative to DMSO according to one tailed Student's paired (B) or unpaired (C) *t*-test. *$p < 0.05$; ns, not significant. Source data are available online for this figure.

