## [Peer Review File · EMBO Molecular Medicine]

Lowering mutant huntingtin by small molecules relieves Huntington's disease symptoms and progression

Anat Bahat, Elad Itzhaki, Benjamin Weiss, Michael Tolmasov, Michael Tsoory, Yael Kuperman, Alexander Brandis, Khriesto Shurrush, and Rivka Dikstein

DOI: [10.15252/emmm.202317983](https://doi.org/10.15252/emmm.202317983)

Corresponding authors: Rivka Dikstein (rivka.dikstein@weizmann.ac.il) , Anat Bahat (anat.bahat@weizmann.ac.il)

Review Timeline:

Submission Date:	11th May 23
Editorial Decision:	8th Aug 23
Revision Received:	9th Oct 23
Editorial Decision:	15th Nov 23
Revision Received:	21st Nov 23
Accepted:	7th Dec 23

Editor: Kelly Anderson

Transaction Report:

8th Aug 2023

Dear Prof. Dikstein,

Thank you for the submission of your manuscript to EMBO Molecular Medicine. We have now heard back from the three referees whom we asked to evaluate your manuscript. As you will see from the reports below, the referees find the topic of your study of potential interest. However, they raise substantial concerns on your work, which should be convincingly addressed in a major revision of the present manuscript.

Although the referees find the study to be of potential interest, they also raise a number of concerns about the conclusiveness of the results and several technical issues. Referee 3 concerns regarding the appropriateness of the model can be overcome if you are to convincingly show that mutant and WT htt are differentially targeted through the use of a specific antibody (see comments by Ref 2). All points should be addressed in a point-by-point response upon submission of the revised version. It would be good to discuss your plan to address the referee concerns and I am available to do so in the coming weeks by email or zoom.

I should remind you that it is EMBO Molecular Medicine policy to allow a single round of revision only and that, therefore, acceptance or rejection of the manuscript will depend on the completeness of your responses included in the next, final version of the manuscript. Revised manuscripts should be submitted within three months of a request for revision; they will otherwise be treated as new submissions, except under exceptional circumstances in which a short extension is obtained from the editor.

I look forward to seeing a revised form of your manuscript as soon as possible. Use this link to login to the manuscript system and submit your revision: <https://embomolmed.msubmit.net/cgi-bin/main.plex>

Yours sincerely,

Kelly

Kelly M Anderson, PhD
Scientific Editor
EMBO Molecular Medicine

We require:

- 1) A .docx formatted version of the manuscript text (including legends for main figures, EV figures and tables). Please make sure that the changes are highlighted to be clearly visible.
- 2) Individual production quality figure files as .eps, .tif, .jpg (one file per figure). For guidance, download the 'Figure Guide PDF' (<https://www.embopress.org/page/journal/17574684/authorguide#figureformat>).
- 3) A .docx formatted letter INCLUDING the reviewers' reports and your detailed point-by-point responses to their comments. As part of the EMBO Press transparent editorial process, the point-by-point response is part of the Review Process File (RPF), which will be published alongside your paper.
- 4) A complete author checklist, which you can download from our author guidelines (<https://www.embopress.org/page/journal/17574684/authorguide#submissionofrevisions>). Please insert information in the checklist that is also reflected in the manuscript. The completed author checklist will also be part of the RPF.
- 5) Please note that all corresponding authors are required to supply an ORCID ID for their name upon submission of a revised manuscript.
- 6) It is mandatory to include a 'Data Availability' section after the Materials and Methods. Before submitting your revision, primary datasets produced in this study need to be deposited in an appropriate public database, and the accession numbers and database listed under 'Data Availability'. Please remember to provide a reviewer password if the datasets are not yet public (see <https://www.embopress.org/page/journal/17574684/authorguide#dataavailability>).

In case you have no data that requires deposition in a public database, please state so in this section. Note that the Data

Availability Section is restricted to new primary data that are part of this study.

7) For data quantification: please specify the name of the statistical test used to generate error bars and P values, the number (n) of independent experiments (specify technical or biological replicates) underlying each data point and the test used to calculate p-values in each figure legend. The figure legends should contain a basic description of n, P and the test applied. Graphs must include a description of the bars and the error bars (s.d., s.e.m.).

8) We would also encourage you to include the source data for figure panels that show essential data. Numerical data should be provided as individual .xls or .csv files (including a tab describing the data). For blots or microscopy, uncropped images should be submitted (using a zip archive if multiple images need to be supplied for one panel). Additional information on source data and instruction on how to label the files are available at

13) Author contributions: the contribution of every author must be detailed in a separate section (before the acknowledgments).

14) A Conflict of Interest statement should be provided in the main text.

15) Every published paper now includes a 'Synopsis' to further enhance discoverability. Synopses are displayed on the journal webpage and are freely accessible to all readers. They include a short stand first (maximum of 300 characters, including space) as well as 2-5 one-sentences bullet points that summarizes the paper. Please write the bullet points to summarize the key NEW findings. They should be designed to be complementary to the abstract - i.e. not repeat the same text. We encourage inclusion

of key acronyms and quantitative information (maximum of 30 words / bullet point). Please use the passive voice. Please attach these in a separate file or send them by email, we will incorporate them accordingly.

***** Reviewer's comments *****

Referee #1 (Comments on Novelty/Model System for Author):

This paper studied the effect of RNA polymerase II targeting small molecules on mHTT mRNA and protein. They tried to overcome some problematic issues in HD treatment. The selective targeting mHTT while sparing wtHTT is critical for HD treatment. So does the brain penetration of these small molecules.

They used most commonly used cell lines and a mouse model in HD research. Although it may need a little more word editing, the paper was well written. The experimental designed was straight forward and the conclusion was reasonable. However, there are some weak points which need to be improved.

Figure 1a if showing Q7 cell not affected will be important.

Figure 1b, how was the protein level of mHTT and wtHTT in Q7 and Q111 cells?

For this experiment if you use the heterozygous cells (Q7/Q111) and confirm that SPIs only reduce mHTT but not wtHTT, that will be more convincing.

For figure 1D and 1E, it will be better if including Q7 cells.

Also, poly Q only detects mHTT. It can't tell wtHTT level. If use antibodies that can recognize both wtHTT and mHTT in heterozygous HD cells, it will strongly support that SPIs sparing wt HTT. This is the same for figure 2D.

Since your novelty is these SPIs only target mHTT, having data indicating that wtHTT is not affected is very important.

Have you check the Nfl in long term treated HD mice?

For your altered genes after treatment, did you see mHTT/wtHTT change?

It will be better to know the protein levels of mHTT and wt HTT in treated mice, figure 3D and 3G.

Referee #2 (Remarks for Author):

This is a nice in vivo followup study to the group's 2019 publication on Spt5-Pol II inhibitors. The idea of mHtt specific is very appealing for HD therapeutics. This work would be greatly strengthened by inclusion of brain PK for all of the administration routes, particularly the ones used for the long-term studies. The limited stability of the compounds should be acknowledged; future studies could work on development of analogs with greater stability.

Fig S1C: Stability does not appear to be robust for 28 days at 37 C (29% degradation -24; 39% degradation -77; 98% -16). Is this the same vehicle used for the in vivo experiments? % DMSO? Similar concentration used in vivo?

Fig S1D: Text and data are inconsistent: text says no significant effect of selected mRNAs with CAG repeats, but several mRNAs did indeed have a statistically significant effect? RNASeq in the cells would be helpful to comment on lack of off-target effects and specificity of the compounds.

Fig 1E- Please specify Ab in text or figure legend.

2C: Consider normalizing to HKG, rather than WT Htt?

2D: Please specify polyQ Ab. The n's per group are 3-8, please state if the assumptions for a parametric t-test were met.

P7 If you want to make the statement that BDNF and mtDNA are HD biomarkers, please include references to their use as biomarkers.

Fig 2E BDNF mRNA levels are extremely low in the striatum (Zuccato references shows mRNA bands were not detectable after amplification of exon I, II or IV mRNAs, only exon III mRNA had a faint band). Did you measure BDNF protein, striatal BDNF protein is robustly decreased in the striatum of various HD models? Text says both SPIs restore BDNF levels, but only SPI-24 is statistically significant.

Fig 2F. Please clarify the age of the mice, this may need a supplemental table. The BACHD with DMSO have a significant reduction in mtND2 over the course of 28 days? Did the WT mice also receive DMSO? The WT + DMSO is an important control to track on DMSO impact on mtND2. What age were the WT mice taken? Please clarify these details in the figure legend/supplemental table. Please include stats between DMSO post vs SPI-24 and SPI-77 post.

Fig 2G-J. Please confirm that these datasets met the assumptions for parametric t-tests. You decided not to proceed with open field since there was no genotypic difference between WT and BACHD, the same appears true for the crossing time and auditory startle tests. Please include stats between DMSO post vs SPI-24 and SPI-77 post.

The Pharmacokinetics of SPI-24 and SPI-77 are missing in Fig 2. Please examine tissue for the presence of SPI-24 and SPI-77 at the end of 28 days to assure these brain regions were exposed to the molecules.

Fig 3B. Is the graph mislabeled for 2 vs 10mg/kg?

Fig 3. The brain PK data after subcutaneous and oral administration is needed.

Fig 3F. Please specify if there are WT mice treated with DMSO vs SPI compounds.

Fig 4A. Please clarify in text or figure legend (or perhaps in methods) whether the fructose was given to both the WT and DMSO groups, and whether the WT and DMSO groups have the same exact water content (fructose and DMSO).

Table S2A Please comment on why the drinking water stability of SPI-24 and -77 was done in water with 25% fructose versus 8% used in experiments. It also appears that the compounds are not very stable in drinking water. The text says gradual decrease, whereas the data suggests more- statistics regarding degradation in this table should be presented here.

Fig 4B-D. Please include the statistical comparison to compare between DMSO post vs SPI-24 and SPI-77 post.

The brain PK data from this study is needed.

Figure 5B. Please include the statistical comparison to compare between DMSO post vs SPI-24 for the beam walk test.

Plasma and brain PK data from this study is needed.

Discussion p13. There is a typo in "significantly reduced the expression level of mutant Htt mRNA in the striatum and increased the level of BDNF and mitochondrial DNA"- DNA was not measured in study

Referee #3 (Comments on Novelty/Model System for Author):

1. Relevance for HD, choice of model systems - Alleles with lower CAGs. The authors state that the tested inhibitors of Spt5 (SPIs) are specifically reducing mutant HTT transcription, while leaving the production from wild-type pretty spared. However, the model systems used (Striatal cells 111 CAGs, human fibroblasts 180 CAGs and BACHD models 97 CAGs) do not faithfully recapitulate the distribution of the human HD alleles, the majority of which has between 40 and 50 CAG expansion. Are the SPIs still effective and specific in the discrimination of 'lower' alleles size?
2. Relevance for HD, choice of model systems - heterozygous conditions. Because of the characteristics of the human HD populations - vast majority are individuals with only one mutant allele - then the SPIs should be tested in heterozygous conditions to specifically prove their efficiency and specificity in cells where the two alleles - wild-type (to be preserved) and mutant (to be reduced) - are present at the same time.
3. Specificity for HTT CAGs. The authors state that SPIs are specifically hitting HTT CAG expansion while other transcripts in the genome, bearing repetitive CAG tracts, remain unaffected. This is a crucial point that needs to be carefully addressed by inspecting the levels of expression of genes/transcripts such as tata-box binding protein (Tbp), androgen receptors and others to exclude possible off-target effects of the treatment. Especially important would be to explore transcripts with polymorphic CAG tracts highly expressed in the brain.

Referee #3 (Remarks for Author):

Bahat and colleagues presents an interesting extension of their previous characterization of novel small molecule inhibitors of Spt5-Pol II, as possible therapeutic strategy in the context of Huntington's Disease pathology.

The paper is testing a relevant hypothesis, proposing a novel strategy to reduce HTT transcriptional levels, utilizing in vivo animal models of the disease. The topic is very central in the HD research community and definitely warrants further dissection and characterization.

Unfortunately, there are 3 crucial hurdles which curb the enthusiasm for this work and challenges the solidity of the authors main findings.

We suggest a specific and precise addressing of these points before this paper can meet the standards for an EMBO Molecular Medicine publication.

Main concerns:

1. Relevance for HD, choice of model systems - Alleles with lower CAGs. The authors state that the tested inhibitors of Spt5 (SPIs) are specifically reducing mutant HTT transcription, while leaving the production from wild-type pretty spared. However, the model systems used (Striatal cells 111 CAGs, human fibroblasts 180 CAGs and BACHD models 97 CAGs) do not faithfully recapitulate the distribution of the human HD alleles, the majority of which has between 40 and 50 CAG expansion. Are the SPIs still effective and specific in the discrimination of 'lower' alleles size?
2. Relevance for HD, choice of model systems - heterozygous conditions. Because of the characteristics of the human HD populations - vast majority are individuals with only one mutant allele - then the SPIs should be tested in heterozygous conditions to specifically prove their efficiency and specificity in cells where the two alleles - wild-type (to be preserved) and mutant (to be reduced) - are present at the same time.
3. Specificity for HTT CAGs. The authors state that SPIs are specifically hitting HTT CAG expansion while other transcripts in the genome, bearing repetitive CAG tracts, remain unaffected. This is a crucial point that needs to be carefully addressed by inspecting the levels of expression of genes/transcripts such as tata-box binding protein (Tbp), androgen receptors and others to exclude possible off-target effects of the treatment. Especially important would be to explore transcripts with polymorphic CAG tracts highly expressed in the brain.

Moderate concern:

1. Transcriptomic analyses - To assess the specificity of the treatment used, the authors resource to RNA-sequencing in order to profile transcriptional changes at genome-wide levels. While this approach is certainly comprehensive and should provide an unbiased view of the possible transcriptomic changes, nevertheless several technicalities limit the solidity of the conclusions. A higher number of reads/sample should be analyzed and possible alternative strategies of profiling - longer reads (100bp) and pair-end sequencing - should be employed. This strategy should enable a better characterization of i. lower abundance transcripts and ii. facilitate the annotation of reads with repetitive stretches.

Because this entire work relates to transcriptional inhibition of repetitive stretches, an additional suggestion would be to use the recently releases T2T-CHM13v1.1 genome assembly which enables to assess centromeric and telomeric highly repetitive genomic regions.

2. Alternative Splicing alteration following SPI treatment. While transcription should clearly be monitored during the SPI treatments, another co-transcriptional process of crucial importance for proper nervous system functioning is alternative splicing. It would be important to test the specificity of the SPIs compounds in the modulation of alternative splicing across different brain areas and at different time points from the beginning of treatment.

Minor points:

1. Introduction_ 'The translated mutant huntingtin protein contains a polyglutamine (PolyQ) stretch that causes progressive degeneration of nerve cells in the brain and dramatically affects the functional and cognitive abilities of the patient (Gil & Rego, 2008).' While the toxicity of CAG expansion can certainly be ascribed to an elongated polyQ stretch, recent advances in the HD field, point to RNA and DNA toxicity that can contribute to the disease. The authors should acknowledge these alternative molecular mechanisms of toxicity in their introduction.
2. Introduction_ 'However, recent clinical trials with ASOs were halted (Kingwell, 2021) perhaps because the ASOs have the potential to reduce the expression of the WT allele known to have important cellular functions (Cattaneo et al, 2001, 2005)'. The failure of the ASOs clinical trials have been associated also to i. treatment of patients at too advanced stages and ii. too high concentration of the drugs. The sentence in the introduction should be revised accordingly.
3. Introduction_ 'Spt5 has multiple interaction domains that direct binding with DNA upstream sequences,.....'. Please revise sentence for inconsistency.
4. Results_ 'SAR-by-catalog' - Explicit the abbreviations or provide an abbreviation list.
5. Results_ 'Reactive compound' - Explain what is the meaning of 'reactive'
6. Results_ Is there a difference between sexes? Male versus females in the behavioural analyses and the molecular ones?
7. Results_ 'ASR-assay' Explicit abbreviation

Response to referees

Referee #1 (Comments on Novelty/Model System for Author):

Figure 1a if showing Q7 cell not affected will be important.

Please note that the Q7 cell analysis was performed with the active compounds that we selected for further studies and for the experiments in the disease model (see Fig. 1B). We now added a scheme describing the steps that lead to the selection of these compounds (Fig. 1A).

Figure 1b, how was the protein level of mHTT and wtHTT in Q7 and Q111 cells?

We now added the protein level of wtHTT in Q7 along with the Q111 (see Fig. 1D) as suggested.

For this experiment if you use the heterozygous cells (Q7/Q111) and confirm that SPIs only reduce mHTT but not wtHTT, that will be more convincing.

Thanks for this suggestion. While we do not have the Q7/Q111 heterozygous cells, we now examined the effect of the compounds on wt and mutant HTT using human HD patient-derived heterozygous cells (see new Fig. 1E-G). The results confirm the differential effect of the SPIs on wt and mutant HTT.

For figure 1D and 1E, it will be better if including Q7 cells.

Also, poly Q only detects mHTT. It can't tell wtHTT level. If use antibodies that can recognize both wtHTT and mHTT in heterozygous HD cells, it will strongly support that SPIs sparing wt HTT. This is the same for figure 2D.

Since your novelty is these SPIs only target mHTT, having data indicating that wtHTT is not affected is very important.

As indicated above, we now added the protein level of wtHTT in Q7 along with the Q111 (see Fig. 1D), as suggested. We also included an analysis of the effect of the compounds on wt and mutant HTT using human HD patient-derived heterozygous cells (see Fig. 1E-G). For 1E, we now show the effect of SPIs on normal human cells that serve as control for the HD patient-derived cells (see Fig. 1G). The results confirm the selective effect of the SPIs on wt and mutant HTT.

Have you check the Nfl in long term treated HD mice?

Unfortunately, we did not have enough plasma from all the treated mice to test this, but this is an excellent suggestion for our future studies.

For your altered genes after treatment, did you see mHTT/wtHTT change?

For technical reasons of read mapping, it was difficult to assign the reads of the Htt gene to either mouse or human gene with sufficient confidence due to their high homology.

It will be better to know the protein levels of mHTT and wt HTT in treated mice, figure 3D and 3G.

We now added the HTT protein analysis to all the mice experiments except oral short-term treatment for which we did not collect brain samples (See new EV3B, EV4B and EV5D).

Referee #2 (Remarks for Author):

This is a nice in vivo followup study to the group's 2019 publication on Spt5-Pol II inhibitors. The idea of mHtt specific is very appealing for HD therapeutics. This work would be greatly strengthened by inclusion of brain PK for all of the administration routes, particularly the ones used for the long-term studies. The limited stability of the compounds should be acknowledged; future studies could work on development of analogs with greater stability.

Thanks for the suggestions. We now added the brain PK of the various administration routes reported in the paper. Direct injection is in EV2C, short SC in EV3C, oral long in EV4A, and SC long in EV5C. We also discuss the limited stability of prolonged delivery (without refreshing) in the discussion section (see paragraph 4 in the Discussion).

Fig S1C: Stability does not appear to be robust for 28 days at 37 C (29% degradation -24; 39% degradation -77; 98% -16). Is this the same vehicle used for the in vivo experiments? % DMSO? Similar concentration used in vivo?

The same concentrations of the compounds and DMSO that were tested in the stability experiment were used for the in vivo studies. These details were now included in the legend of Table EV1.

Fig S1D: Text and data are inconsistent: text says no significant effect of selected mRNAs with CAG repeats, but several mRNAs did indeed have a statistically significant effect? RNASeq in the cells would be helpful to comment on lack of off-target effects and specificity of the compounds.

We thank the referee for this justified comment. This has now been corrected (now Fig. EV1B). As for the off-target effects, we performed RNA-seq following a short SC injection in the animal and found a very limited number of affected genes in response to SPI-24.

Fig 1E- Please specify Ab in text or figure legend.

The antibodies used are marked in the figure, indicated in the legend and their specific details are provided in the Material and Methods.

2C: Consider normalizing to HKG, rather than WT Htt?

Unfortunately, we did not have enough mRNA left to perform the suggested normalization.

2D: Please specify polyQ Ab. The n's per group are 3-8, please state if the assumptions for a parametric t-test were met.

The antibodies are specified in the Material and Methods and indicated in the figure legend. The data in each comparison group show a Normal (Gaussian) distribution according to 4 tests: Anderson-Darling (A2*), D'Agostino-Pearson omnibus (K2), Shapiro-Wilk (W), and Kolmogorov-Smirnov (distance). In addition, the data in each comparison group exhibit similar degrees of homogeneity of variance (Homoscedasticity) according to F test. All tests were performed using PRISM.

P7 If you want to make the statement that BDNF and mtDNA are HD biomarkers, please include references to their use as biomarkers.

We added the relevant reference stating BDNF as a biomarker (see p. 7, second paragraph). The references for mtDNA are already mentioned in the text.

Fig 2E BDNF mRNA levels are extremely low in the striatum (Zuccato references shows mRNA bands were not detectable after amplification of exon I, II or IV mRNAs, only exon III mRNA had a faint band). Did you measure BDNF protein, striatal BDNF protein is robustly decreased in the striatum of various HD models? Text says both SPIs restore BDNF levels, but only SPI-24 is statistically significant.

We didn't measure BDNF protein level. According to Gray M. et al (2008) there is a significant reduction of BDNF mRNA expression in the brain of BACHD mice relative to wild type. The text, which describes the SPIs effect on the BDNF mRNA level, was modified (both SPIs increase the level of BDNF: the effect of SPI-24 is significant and the p-value for SPI-77 is 0.06). (see p. 7, second paragraph).

Fig 2F. Please clarify the age of the mice, this may need a supplemental table.

We thank the referee for his/her suggestion and we now added a table summarizing all the mice used in this study, including their age, genotyping and treatment (Appendix table S1).

The BACHD with DMSO have a significant reduction in mtND2 over the course of 28 days?
As expected from the disease progression, without any treatment (DMSO group) there is a reduction in mtND2 level over the course of 28 days. This is now clarified in the text (p. 8, first paragraph).
Did the WT mice also receive DMSO? The WT + DMSO is an important control to track on DMSO impact on mtND2. What age were the WT mice taken? Please clarify these details in the figure legend/supplemental table.

We now clarified throughout the text, the figure legends, and in the Material and Methods that the WT mice were untreated and are from the same littermate of the treated BACHD mice.

Please include stats between DMSO post vs SPI-24 and SPI-77 post.

Thanks for the suggestion, we now included the statistics between DMSO post vs SPI-24 and SPI-77 post for Figure 2F and the differences were found significant.

Fig 2G-J. Please confirm that these datasets met the assumptions for parametric t-tests.

We verified that the datasets for figures 2G-J met the assumptions for parametric t-tests. For Figure 2G and 2H, the t-test between WT and BACHD DMSO was corrected with Welch t-test since the populations have different standard deviations.

You decided not to proceed with open field since there was no genotypic difference between WT and BACHD, the same appears true for the crossing time and auditory startle tests.

This is true that the crossing time between WT and BACHD is not statistically significant. However, this parameter does not stand on its own but rather complements the paw steps number and slip ratio parameters that differ significantly between WT and BACHD. Regarding the auditory startle test, there is an apparent phenotypic change between WT which displays a significant difference between B1 and B3, and BACHD which has no difference between B1 and B3.

Please include stats between DMSO post vs SPI-24 and SPI-77 post.

We now included the statistics between DMSO post vs SPI-24 and SPI-77 post for Figure 2H.

The Pharmacokinetics of SPI-24 and SPI-77 are missing in Fig 2. Please examine tissue for the presence of SPI-24 and SPI-77 at the end of 28 days to assure these brain regions were exposed to the molecules.

We thank the referee for this suggestion. We now added the brain PK of the various administration routes reported in the paper. Direct injection is in EV2C, short SC in EV3C, oral long in EV4A and SC long in EV5C.

Fig 3B. Is the graph mislabeled for 2 vs 10mg/kg?

We thank the referee for noting the error. It is now corrected.

Fig 3. The brain PK data after subcutaneous and oral administration is needed.

The brain PK data for the short-term SC treatment are shown in EV3C. Unfortunately, we did not collect brain samples from the oral short-term treatment (except for RNA analysis).

Fig 3F. Please specify if there are WT mice treated with DMSO vs SPI compounds.

We add clarifications throughout the text and in the material and methods that the WT mice were left untreated.

Fig 4A. Please clarify in text or figure legend (or perhaps in methods) whether the fructose was given to both the WT and DMSO groups, and whether the WT and DMSO groups have the same exact water content (fructose and DMSO).

As mentioned above, only BACHD mice were treated. The same concentration of fructose was given to all treated groups (DMSO and SPIs).

Table S2A Please comment on why the drinking water stability of SPI-24 and -77 was done in water with 25% fructose versus 8% used in experiments. It also appears that the compounds are not very

stable in drinking water. The text says gradual decrease, whereas the data suggests more- statistics regarding degradation in this table should be presented here.

Indeed, the initial stability test was done in water containing 25% fructose, which is the concentration we expected to add in order for the mice to drink the SPIs-containing water. In practice, in preliminary tests, we found that 8% is sufficient in order to drink the SPIs-containing water. We added the statistics to Table EV2A and corrected the relevant text.

Fig 4B-D. Please include the statistical comparison to compare between DMSO post vs SPI-24 and SPI-77 post.

For panel B, the change between B3 to B1 is the important parameter for this assay. In panel C, the ratio between post and pre within each group is the most important measure. We added the statistics of DMSO post vs SPI-24 and SPI-77 post to panel D.

The brain PK data from this study is needed.

The brain PK data for the long-term oral treatment were performed and shown in EV4A.

Figure 5B. Please include the statistical comparison to compare between DMSO post vs SPI-24 for the beam walk test.

This statistic was found significant and added.

Plasma and brain PK data from this study is needed.

The brain PK data for the long-term SC treatment are shown in EV5C.

Discussion p13. There is a typo in "significantly reduced the expression level of mutant Htt mRNA in the striatum and increased the level of BDNF and mitochondrial DNA"- DNA was not measured in study

Corrected.

Referee #3 (Remarks for Author):

Main concerns:

1. Relevance for HD, choice of model systems - Alleles with lower CAGs. The authors state that the tested inhibitors of Spt5 (SPIs) are specifically reducing mutant HTT transcription, while leaving the production from wild-type pretty spared. However, the model systems used (Striatal cells 111 CAGs, human fibroblasts 180 CAGs and BACHD models 97 CAGs) do not faithfully recapitulate the distribution of the human HD alleles, the majority of which has between 40 and 50 CAG expansion. Are the SPIs still effective and specific in the discrimination of 'lower' alleles size?

2. Relevance for HD, choice of model systems - heterozygous conditions. Because of the characteristics of the human HD populations - vast majority are individuals with only one mutant allele - then the SPIs should be tested in heterozygous conditions to specifically prove their efficiency and specificity in cells where the two alleles - wild-type (to be preserved) and mutant (to be reduced) - are present at the same time.

We thank the referee for these important comments. To test the effect of the SPIs on HD alleles bearing CAG expansion numbers that recapitulate the distribution in humans, we employed HD patient-derived heterozygous cells bearing 44, 55, and 66 CAG repeats along with normal Htt allele. The effects of the SPIs on the wt and mut alleles were tested using western blot and are presented in Figure 1E-G.

3. Specificity for HTT CAGs. The authors state that SPIs are specifically hitting HTT CAG expansion while other transcripts in the genome, bearing repetitive CAG tracts, remain unaffected. This is a crucial point that needs to be carefully addressed by inspecting the levels of expression of

genes/transcripts such as tata-box binding protein (Tbp), androgen receptors and others to exclude possible off-target effects of the treatment. Especially important would be to explore transcripts with polymorphic CAG tracts highly expressed in the brain.

The specificity for HTT CAG was shown in Figure EV1B. We now added to this figure the SPIs effect on AR gene.

Moderate concern:

1. Transcriptomic analyses - To assess the specificity of the treatment used, the authors resource to RNA-sequencing in order to profile transcriptional changes at genome-wide levels. While this approach is certainly comprehensive and should provide an unbiased view of the possible transcriptomic changes, nevertheless several technicalities limit the solidity of the conclusions. A higher number of reads/sample should be analyzed and possible alternative strategies of profiling - longer reads (100bp) and pair-end sequencing - should be employed. This strategy should enable a better characterization of i. lower abundance transcripts and ii. facilitate the annotation of reads with repetitive stretches.

Because this entire work relates to transcriptional inhibition of repetitive stretches, an additional suggestion would be to use the recently releases T2T-CHM13v1.1 genome assembly which enables to assess centromeric and telomeric highly repetitive genomic regions.

We consulted with a bioinformatics expert at the Weizmann Institute before conducting the deep sequencing analysis and he pointed out that 40 million reads/sample are considered excellent depth (See also: Baccarella, A., Williams, C.R., Parrish, J.Z. et al. Empirical assessment of the impact of sample number and read depth on RNA-Seq analysis workflow performance. BMC Bioinformatics 19, 423; 2018). Furthermore, the consensus in the field supports the acceptability of employing 50 bp reads for the accurate determination of gene expression levels through alignment.

Although we studied the human Htt gene, it was transformed into mice and studied in the mouse genome context. Therefore, although very nice, the T2T-CHM13v1.1 human assembly was not suitable.

2. Alternative Splicing alteration following SPI treatment. While transcription should clearly be monitored during the SPI treatments, another co-transcriptional process of crucial importance for proper nervous system functioning is alternative splicing. It would be important to test the specificity of the SPIs compounds in the modulation of alternative splicing across different brain areas and at different time points from the beginning of treatment.

Although exploring the impact of SPIs on alternative splicing is beyond the scope of this manuscript, and existing literature indicates that Spt4/Spt5 have not been implicated in alternative splicing, we tested the SPIs effect on HD-specific alternatively-spliced genes reported in Elorza et al., 2021, by qPCR. The results, showing no effect of both SPIs on alternative splicing of these genes, are presented in Appendix figure S3.

Minor points:

1. Introduction_ 'The translated mutant huntingtin protein contains a polyglutamine (PolyQ) stretch that causes progressive degeneration of nerve cells in the brain and dramatically affects the functional and cognitive abilities of the patient (Gil & Rego, 2008).' While the toxicity of CAG expansion can certainly be ascribed to an elongated polyQ stretch, recent advances in the HD field, point to RNA and DNA toxicity that can contribute to the disease. The authors should acknowledge these alternative molecular mechanisms of toxicity in their introduction.

We thank the referee for this clarification. This was now added to the introduction (see p. 3, first paragraph).

2. Introduction_ 'However, recent clinical trials with ASOs were halted (Kingwell, 2021) perhaps because the ASOs have the potential to reduce the expression of the WT allele known to have important cellular functions (Cattaneo et al, 2001, 2005)'. The failure of the ASOs clinical trials have

been associated also to i. treatment of patients at too advanced stages and ii. too high concentration of the drugs. The sentence in the introduction should be revised accordingly.

We revised the sentence as suggested (see p. 3 second paragraph).

3. Introduction_ 'Spt5 has multiple interaction domains that direct binding with DNA upstream sequences,.....'. Please revise sentence for inconsistency.

The sentence was revised.

4. Results_ 'SAR-by-catalog' - Explicit the abbreviations or provide an abbreviation list.

The abbreviation is now explicit in the text and in the material and methods.

5. Results_ 'Reactive compound' - Explain what is the meaning of 'reactive'

'Reactive compound' is defined as a compound that easily undergoes a chemical reaction, either by itself or with other materials. This is now clarified in the text.

6. Results_ Is there a difference between sexes? Male versus females in the behavioural analyses and the molecular ones?

No significant differences were observed between the sexes. This is now added to the material and methods.

7. Results_ 'ASR-assay' Explicit abbreviation

The abbreviation is now explicit in the text and in the material and methods.

Dear Rivka,

Congratulations on a great revision! Overall, the referees have been positive. While referee 3 has two remaining minor concerns, we will leave it to your discretion as to whether you will address the concerns by further experiments or added discussion.

When you submit your revised version, please also take care of the following editorial items and add this also to your point-by-point response:

1. Please remove the author contribution section from the main manuscript.
2. Please update the appendix file to include page numbers.
3. We require the publication of source data, particularly for electrophoretic gels and blots and graphs, with the aim of making primary data more accessible and transparent to the reader. Would you please provide me with a PDF file per figure that contains the original, uncropped and unprocessed scans of all or key gels used in the figure or for graphs, an Excel spreadsheet with the original data used to generate the graphs. The PDF files should be labeled with the appropriate figure/panel number, and should have the molecular weight marker. The PDF files will be published online with the article as supplementary 'Source Data'.
4. Please adjust the synopsis image size to 550 pixels wide by 200-400 pixels high.
5. Please upload Table 1 in an editable format, no mouse image as it will be typeset. Alternatively, it can be made into an EV table, but then it would need its legend added to the file.
6. Would you consider combining Table EV2A and B into one file. EV tables need their legends removed from the manuscript and added to the tables.
7. The figure legend style does not comply with our guidelines, please review our format in the author guidelines listed online. For example, a 'Date Information' statement is required in each legend.
8. Please define the annotated p values ****/**/*/* in the legend of figure 1a-e, g; 2c-j; 3c-e, g-h; 4b, d; 5b-c; EV1b.
9. Please note that the box plots need to be defined in terms of minima, maxima, centre, bounds of box and whiskers, and percentile in the legend of figures 2c-i; 3d, e, g; 4c; 5b; EV3d.
10. Please note that the box plots need to be defined in terms of minima, maxima, centre, bounds of box, and percentile in the legend of figures 3h.
11. Please define the error bars in the legend of figure EV3a.

Thank you for the opportunity to consider your work for publication, I look forward to your revision.

Warm wishes,
Kelly

Kelly M Anderson, PhD
Scientific Editor
EMBO Molecular Medicine

*** Instructions to submit your revised manuscript ***

When submitting your revised manuscript, please

include:

1) a .docx formatted version of the manuscript text (including Figure legends and tables)

2) Separate figure files*

3) supplemental information as Expanded View and/or Appendix. Please carefully check the authors guidelines for formatting Expanded view and Appendix figures and tables at <https://www.embopress.org/page/journal/17574684/authorguide#expandedview>

4) a letter INCLUDING the reviewer's reports and your detailed responses to their comments (as Word file).

5) The paper explained: EMBO Molecular Medicine articles are accompanied by a summary of the articles to emphasize the major findings in the paper and their medical implications for the non-specialist reader. Please provide a draft summary of your article highlighting

6) For more information: There is space at the end of each article to list relevant web links for further consultation by our readers. Could you identify some relevant ones and provide such information as well? Some examples are patient associations, relevant databases, OMIM/proteins/genes links, author's websites, etc...

7) Author contributions: the contribution of every author must be detailed in a separate section.

8) EMBO Molecular Medicine now requires a complete author checklist (<https://www.embopress.org/page/journal/17574684/authorguide>) to be submitted with all revised manuscripts. Please use the checklist as guideline for the sort of information we need WITHIN the manuscript. The checklist should only be filled with page numbers where the information can be found. This is particularly important for animal reporting, antibody dilutions (missing) and exact values and n that should be indicated instead of a range.

9) Every published paper now includes a 'Synopsis' to further enhance discoverability. Synopses are displayed on the journal webpage and are freely accessible to all readers. They include a short stand first (maximum of 300 characters, including space) as well as 2-5 one sentence bullet points that summarise the paper. Please write the bullet points to summarise the key NEW findings. They should be designed to be complementary to the abstract - i.e. not repeat the same text. We encourage inclusion of key acronyms and quantitative information (maximum of 30 words / bullet point). Please use the passive voice. Please attach these in a separate file or send them by email, we will incorporate them accordingly.

You are also welcome to suggest a striking image or visual abstract to illustrate your article. If you do please provide a jpeg file 550 px-wide x 300-800px high.

10) A Conflict of Interest statement should be provided in the main text

11) Please note that we now mandate that all corresponding authors list an ORCID digital identifier. This takes <90 seconds to complete. We encourage all authors to supply an ORCID identifier, which will be linked to their name for unambiguous name identification.

Currently, our records indicate that the ORCID for your account is 0000-0002-6251-4723.

Please click the link below to modify this ORCID:
Link Not Available

Each figure should be given in a separate file and should have the following resolution:
Graphs 800-1,200 DPI

Photos 400-800 DPI
Colour (only CMYK) 300-400 DPI"

*Additional important information regarding figures and illustrations can be found at <https://bit.ly/EMBOPressFigurePreparationGuideline>. See also figure legend preparation guidelines: <https://www.embopress.org/page/journal/17574684/authorguide#figureformat>

***** Reviewer's comments *****

Referee #1 (Comments on Novelty/Model System for Author):

An adequate method was used to study the wt HTT VS mHTT level. They tried to overcome some problematic issues in HD treatment. Selective targeting mHTT while sparing wtHTT is critical for HD treatment. So does the brain penetration of these small molecules. They used the most commonly used cell lines and a mouse model in HD research.

Referee #1 (Remarks for Author):

The author correctly addressed all the minor issues I was concerned about.

Referee #3 (Comments on Novelty/Model System for Author):

This study represents an interesting extension of their previous characterization of novel small molecule inhibitors of Spt5-Pol II, as possible therapeutic strategy in the context of Huntington's Disease pathology. The paper is testing a relevant hypothesis, proposing a novel strategy to reduce HTT transcriptional levels, utilizing in vivo animal models of the disease. The topic is very central in the HD research community and definitely warrants further dissection and characterization.

Referee #3 (Remarks for Author):

In general, the authors addressed most of my previous comments appropriately.

A couple of concerns remain:

1. In the newly inserted HD patients' human cell lines, with different CAG and in heterozygous conditions, the quantification of mutant and wild-type huntingtin could be done using an antibody (i.e MAB 2166) able to distinguish both proteins, and, probing the same membrane. Especially for longer CAG tracts, a better separation of the wild-type and mutant alleles, might be achieved;
2. The specificity of the treatment for HTT-CAG repeat was tested by inspecting the levels of AR, possibly having included a broader array of CAG-containing transcripts, might have strengthened the conclusions.

The authors addressed the minor editorial issues.

7th Dec 2023

Dear Rivka,

Congratulations on an excellent manuscript, I am pleased to inform you that your manuscript has been accepted for publication in The EMBO Journal. Thank you for your comprehensive response to the referee concerns and for providing detailed source data. It has been a pleasure to work with you to get this to the acceptance stage.

I will begin the final checks on your manuscript before submitting to the publisher next week. Once at the publisher, it will take about 3 weeks for your manuscript to be published online. As a reminder, the entire review process, including referee concerns and your point-by-point response, will be available to readers.

I will be in touch throughout the final editorial process until publication. In the meantime, I hope you find time to celebrate!

Warm wishes,
Kelly

Kelly M Anderson, PhD
Scientific Editor
EMBO Molecular Medicine

Please note that you will be contacted by Springer Nature Author Services to complete licensing and payment information.
